# Physical observables to determine the nature of membrane-less cellular sub-compartments

**Mathias L Heltberg[1,2], Judith Miné-Hattab[2], Angela Taddei[2], Aleksandra M Walczak[1]\*, Thierry Mora[1]\***

[1]Laboratoire de physique de l'École normale supérieure, CNRS, PSL University, Sorbonne Université, and Université de Paris, Paris, France; [2]Institut Curie, CNRS, PSL University, Sorbonne Université, Paris, France

**Abstract** The spatial organization of complex biochemical reactions is essential for the regulation of cellular processes. Membrane-less structures called foci containing high concentrations of specific proteins have been reported in a variety of contexts, but the mechanism of their formation is not fully understood. Several competing mechanisms exist that are difficult to distinguish empirically, including liquid-liquid phase separation, and the trapping of molecules by multiple binding sites. Here, we propose a theoretical framework and outline observables to differentiate between these scenarios from single molecule tracking experiments. In the binding site model, we derive relations between the distribution of proteins, their diffusion properties, and their radial displacement. We predict that protein search times can be reduced for targets inside a liquid droplet, but not in an aggregate of slowly moving binding sites. We use our results to reject the multiple binding site model for Rad52 foci, and find a picture consistent with a liquid-liquid phase separation. These results are applicable to future experiments and suggest different biological roles for liquid droplet and binding site foci.

**\*For correspondence:**
aleksandra.walczak@phys.ens.fr (AMW);
thierry.mora@gmail.com (TM)

## Editor's evaluation

There has been a lively debate recently concerning the multiplicity of reported observations of phase-separated compartments inside of cells. Specifically, some claims of phase separation have been challenged, and an alternative model has been put forward that explains clustering of observed particles as resulting from colocalization of binding sites with no phase separation. The current study does an admirable job of proposing and analyzing ways of distinguishing these two scenarios.

## Introduction

The cell nucleus of eukaryotic cells is not an isotropic and homogeneous environment. In particular, it contains membrane-less sub-compartments, called foci or condensates, where the protein concentration is enhanced for certain proteins. Even though foci in the nucleus have been observed for a long time, the mechanisms of their formation, conservation, and dissolution are still debated (*Strom et al., 2017*; *Altmeyer et al., 2015*; *Larson et al., 2017*; *Patel et al., 2015*; *Boehning et al., 2018*; *Pessina et al., 2019*; *McSwiggen et al., 2019b*; *McSwiggen et al., 2019a*; *Oshidari et al., 2020*; *Gitler et al., 2020*; *Erdel et al., 2020*). An important aspect of these sub-compartments is their ability to both form at the correct time and place, and also to dissolve after a certain time. One example of foci are the structures formed at the site of a DNA double strand break (DSB) in order to localize vital

**Figure 1.** Schematic setup of the models. (**A**) In the middle, the observed signal from a fluorescently tagged Rad52 protein inside the nucleus following a double-stand break. Left: Schematic figure showing the Polymer Bridging Model (PBM). Proteins binding specifically to the chromatin stabilize it, effectively trapping the motion of other molecules. Right: Schematic figure showing the Liquid Phase Model (LPM). Liquid-liquid phase separation results in the formation of a droplet foci with a different potential and different effective diffusion properties than outside the droplet. (**B**) Details of the PBM model. Particles diffuse freely with diffusivity $D_n$ until they hit one of the $N$ spherical binding sites, themselves diffusing with diffusivity $D_b$. The focus is formed due a high concentration of binding sites. The binding sites are only partially absorbing, so that not all collision events result in a binding even. Once bound, the particle stays attached to the binding site, and then unbinds with rate $k_-$.

proteins for the repair process at the site of a DNA break (*Lisby et al., 2001*). Condensates have also been reported to be involved in gene regulation (*Hnisz et al., 2017*; *Bing et al., 2020*) and in the grouping of telomeres in yeast cells (*Meister and Taddei, 2013*; *Ruault et al., 2021*). More generally, a vast number of membrane-less cellular sub-compartments that have been reported in the literature with different names. Here, we consider a focus to be a spherical condensate of size smaller than a few hundreds nanometers.

Different hypotheses have been put forward to explain focus formation in the context of chromatin, among which two main ones (discussed in the particular context of DSB foci in *Miné-Hattab and Taddei, 2019*): the Polymer Bridging Model (PBM) and the Liquid Phase Model (LPM). The Polymer Bridging Model is based on the idea that specific proteins form bridges between different chromatin loci by creating loops or by stabilizing interactions between distant loci on the DNA (*Figure 1A*, left). These interactions can be driven by specific or multivalent weak interactions between chromatin binding proteins and chromatin components. In this case, the existence of sub-compartments relies on both the binding and bridging properties of these proteins. By contrast, the LPM posits that membrane-less sub-compartments arise from a liquid-liquid phase separation. In this picture,

first proposed for P granules involved in germ cell formation (*Brangwynne et al., 2009*), proteins self-organize into liquid-like spherical droplets that grow around the chromatin fiber, allowing certain molecules to become concentrated while excluding others (*Figure 1A*, right).

Although some biochemical and wide field microscopy data support the LPM hypothesis for DSB foci (*Altmeyer et al., 2015*; *Larson et al., 2017*; *Strom et al., 2017*; *McSwiggen et al., 2019b*), these observations are at the optical resolution limit, and a more direct detection of these structures is still missing. Coarse-grained theoretical models of the LPM exist (*Statt et al., 2020*; *Grmela and Öttinger, 1997*), but predictions of microscale behavior that can be combined with a statistical analysis of high-resolution microscopy data to discriminate between the hypotheses has not yet been formulated. Previously, we analyzed in detail single-particle tracking data in the context of yeast DSB foci (*Miné-Hattab et al., 2021*). We found that the behavior of Rad52 foci was consistent with a liquid droplet, based on several observations, including the diffusion coefficient of proteins inside the focus relative to that of the whole focus, the size of the focus following two double-strand breaks, and its dissolution upon adding aliphatic alcohol hexanediol.

Here, we build a general physical framework for understanding and predicting the behavior of each model under different regimes. The framework is general and applicable to many different types of foci, although we chose to focus on the regime of parameters relevant to yeast DSB foci, for which we can directly related our results to experimental measurements. While the LPM and PBM models have often been presented in the literature as opposing views, here we show under what conditions the PBM may be reduced to an effective description that is mathematically equivalent to the LPM, but with specific constraints linking its properties. We discuss the observables of the LPM and PBM and derive features that can be used to discriminate these two scenarios.

## Results
### Two models of foci

To describe the situation measured in single particle tracking experiments, we consider the diffusive motion of a single molecule within the nucleus of a cell in the overdamped limit, described by the Langevin equation in three dimensions (using the Itô convention, as we will for the rest of this work):

$$d\mathbf{r} = dt\left[\nabla D(\mathbf{r}) - \frac{D(\mathbf{r})}{k_B T}\nabla U(\mathbf{r})\right] + \sqrt{2D(\mathbf{r})}\,d\mathbf{W}, \tag{1}$$

where $\mathbf{W}$ is a three-dimensional Wiener process, $U(\mathbf{r})$ is the potential exerted on the particle, and $D(\mathbf{r})$ is a position-dependent diffusion coefficient. The $\nabla U$ term corresponds to a force divided by the drag coefficient $k_B T/D(\mathbf{r})$, which is given in terms of $D$ and temperature according to Einstein's relation. The $\nabla D$ term comes from working within the Itô convention. The steady state distribution of particles is given by the Boltzmann distribution:

$$p(\mathbf{r}) = \frac{1}{Z}\exp\left[-\frac{U(\mathbf{r})}{k_B T}\right], \tag{2}$$

where $Z$ is a normalization constant.

In the LPM, we associate the focus with a liquid droplet characterized by a sudden change in the energy landscape. We model the droplet as a change in the potential $U(r)$, and a change in the diffusion coefficient $D(r)$ inside the droplet focus compared to the diffusion coefficient in the rest of the nucleus $D_n$. We assume both the diffusion coefficient and the potential are spherically symmetric around the center of the focus, and have sigmoidal forms:

$$D(r) = D_0 + \frac{D_n - D_0}{1 + e^{-b(r - r_f)}}, \tag{3}$$

$$U(r) = \frac{A}{1 + e^{-b(r - r_f)}}, \tag{4}$$

where $D_0$ is the diffusion coefficient inside the focus, $r_f$ is the radial distance to the center of the focus, and the coefficients are defined in *Table 1*. Different relations between the diffusion coefficient and the surface potential are possible.

**Table 1.** Parameters used in this study with their typical values, and the ranges we have considered. Experimental values are from *Miné-Hattab et al., 2021* (see Materials and Methods for details on estimating diffusion coefficients and free energy differences). $D_0$ and $A$ are model parameters in the LPM, but also effective observables in the PBM. The diffusivity of binding sites is taken to be that of Rfa1 molecules in the focus, which bind to single-stranded DNA in repair foci, and are thus believe to follow the diffusion of the chromatin (*Miné-Hattab et al., 2021*). The number $N$ of binding site is related to their density $\rho$ inside the focus through $N = (4/3)\pi\rho r_f^3$.

| Variable | Model | Description | Value | Range | Exp. value | Units |
|---|---|---|---|---|---|---|
| $r_f$ | both | radius of focus | 100 | 50–200 | | nm |
| $r_n$ | both | radius of nucleus | 500 | 300–1000 | | nm |
| $D_n$ | both | Diffusion coefficient in nucleus | 1.0 | 0.5–2.0 | 1.08 | $\mu m^2/s$ |
| $\sigma$ | both | Experimental noise level | 30 | 30 | 30 | nm |
| $D_0$ | LPM | Diffusion coefficient inside droplet | 0.05 | 0.01–0.5 | 0.032 | $\mu m^2/s$ |
| $A$ | LPM | Surface potential | 5.0 | 0–10 | 5.5 | $k_B T$ |
| $b$ | LPM | Steepness in potential | 1,000 | 500–10000 | | $\mu m^{-1}$ |
| $\rho$ | PBM | Density of binding sites inside focus | $4.8 \cdot 10^4$ | $1 \cdot 10^3$ -$8.4 \cdot 10^4$ | | $\mu m^{-3}$ |
| $D_b$ | PBM | Diffusion coefficient of binding sites | 0.005 | 0 -0.1 | 0.005 | $\mu m^2 s^{-1}$ |
| $r_b$ | PBM | Radius of binding sites | 10 | 5 -20 | | nm |
| $k_-$ | PBM | Unbinding rate | 500 | 10 -10,000 | | $s^{-1}$ |
| $\kappa$ | PBM | Absorption parameter | 100 | 0 -1000 | | $\mu m/s$ |

In the PBM, we describe the dynamics of particles using a model where the focus has $N$ binding sites, each of which is a partially reflecting sphere (*Bryan, 1891*; *Duffy, 2015*; *Carslaw and Jaeger, 1992*) with radius $r_b$ (*Figure 1B and C*). Binding sites can themselves diffuse with diffusion coefficient $D_b$, and are confined within the focus by a potential $U_b(\mathbf{r})$, so that their density is $\rho(\mathbf{r}) \propto e^{-U_b(\mathbf{r})}$ according to the Boltzmann distribution. While not bound, particles diffuse freely with diffusion constant $D_n$, even when inside the focus. However, the movement of the particle is affected by direct interactions with the binding sites. Binding is modeled as follows. As the particle crosses the spherical boundary of a binding site during an infinitesimal time step $dt$, it gets absorbed with probability $p_b = \kappa\sqrt{\pi dt/D_n}$ (*Figure 1C*), where $\kappa$ is an absorption parameter consistent with the Robin boundary condition at the surface of the spheres, $D\mathbf{n} \cdot \nabla p(\mathbf{x}) = \kappa p(\mathbf{x})$ (*Erban and Chapman, 2007*; *Singer et al., 2008*), where $\mathbf{x}$ is a point on the surface of the sphere, and $\mathbf{n}$ is the unit vector normal to it.

While bound, particles follow the motion of their binding site, described by:

$$d\mathbf{r} = -dt\frac{D_b}{k_B T}\nabla U_b(\mathbf{r}) + \sqrt{2D_b}d\mathbf{W}, \tag{5}$$

where $\mathbf{W}$ is a three-dimensional Wiener process. A bound particle is released with a constant rate $k_-$. Since the potential $U_b$ is constant within the bulk and its only function is to keep binding sites within the focus, the PBM can be described by five parameters: $N$, $r_b$, $D_b$, $\kappa$ and $k_-$. Their typical values can be found in *Table 1*.

## Comparison between simulated and experimental traces

In recent experimental work (*Miné-Hattab et al., 2021*), we used single particle tracking to follow the movement of Rad52 molecules, following a double-strand break in *S. cerevisiae* yeast cells, which causes the formation of a focus. These experiments show that temporal traces of Rad52 molecules concentrate inside the focus, as shown for a representative cell in *Figure 2A*.

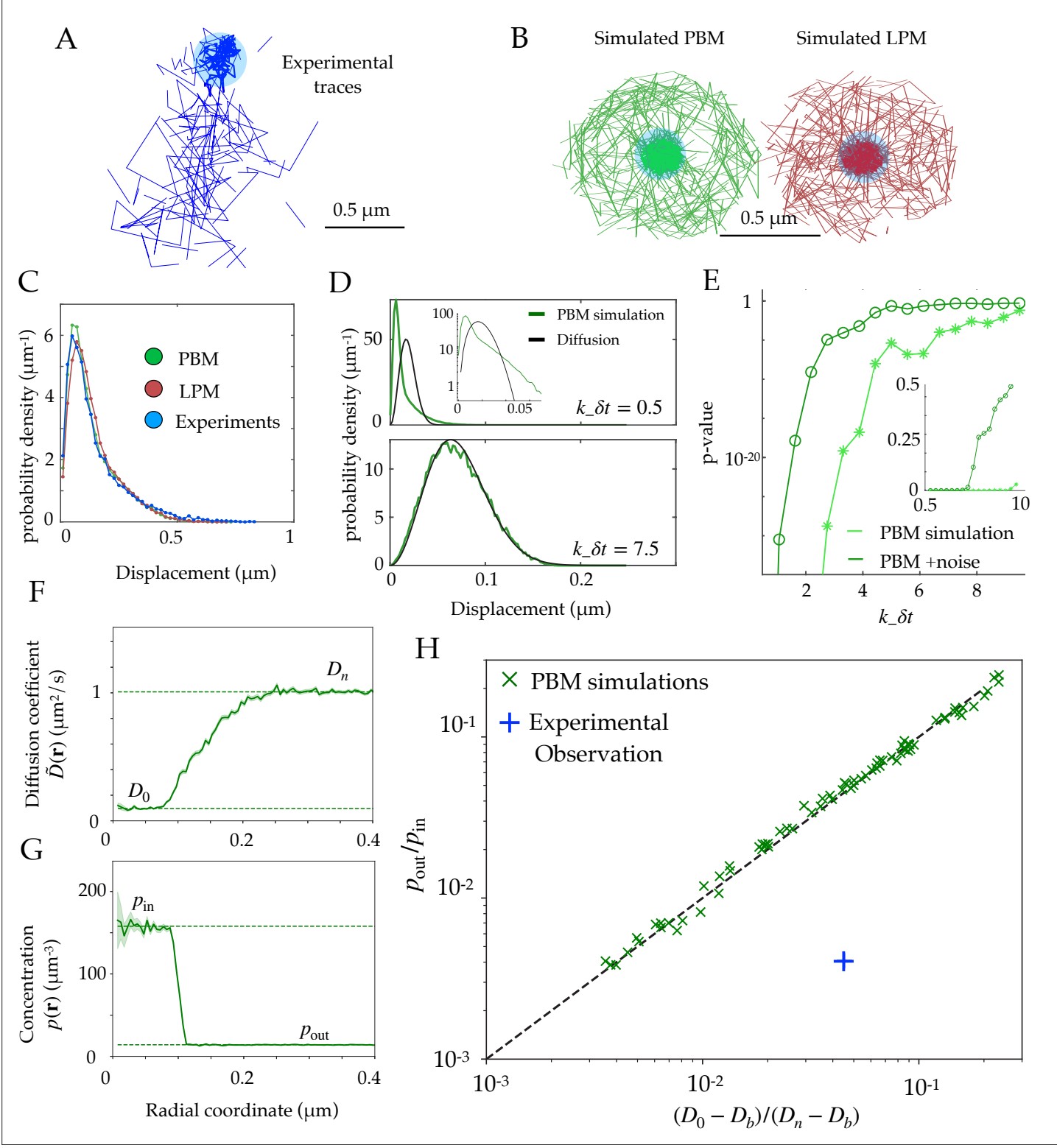

**Figure 2.** Diffusion properties and effective free energy. (**A**) Example of experimental tracking of Rad52 molecules visiting a double-strand break (DSB) focus. Different connected traces correspond to distinct Rad52 molecules. (**B**) Example trajectory of a particle visiting the focus from simulations in the PBM (left) and LPM (right). The simulated trajectories are visually similar to the data in B. (**C**) Displacement histogram (jump sizes) for the PBM, LPM and experiments, for an interval $\delta = 20$ ms. D. Displacement histogram for the PBM for small values of $k\_dt$ (top) and high values (bottom). Here we varied the interval from $\delta t = 1$ ms (top) to $\delta t = 15$ ms (bottom). (**E**) Hypothesis testing using a two sided KS-test, comparing the displacement histogram

*Figure 2 continued on next page*

*Figure 2 continued*

of a free diffusion process (black line in D) and the displacement histogram of diffusion inside the focus (green line in D). Parameters are the same as in (**D**) $\delta t$ was varied from 1 to 25 ms. (**F**) Effective diffusion coefficient as a function of distance to the focus center $r$, estimated from simulations of the PBM calculating $\tilde{D} = \langle \delta r^2 \rangle/(2d\delta t)$ in each radial segment. (**G**) Particle density $p(\mathbf{r})$ as a function of $r$, estimated from simulations of the PBM. Error bars are standard errors on the mean. (**H**) Relation between the ratio $(D_0 - D_b)/(D_n - D_b)$ versus the ratio of densities inside and outside the focus (From Equation 11), both estimated from simulations of the PBM (green crosses), compared to the identity prediction (Equation 12, black line). Blue cross shows the experimental observation for Rad52 in DSB loci (see text related to **Figure 7** and Table S1 in **Miné-Hattab et al., 2021**). Parameter values as in Table 1 except: $r_n = 1 \mu$m for B; $A = 2.5 k_B T$ for B-C; $r_n = 0.3 \ \mu$m, $r_f = 0.15 \ \mu$m and $D_n = 0.5 \ \mu$m$^2$/s for D-E, $\kappa = 300 \ \mu m/s$ for D, $r_n = 0.75 \ \mu$m for F-H. In H we varied $\kappa = 1$–$400 \ \mu m/s$, $k_- = 1$–$1,500$ s-1, and $\rho = 2.4$–$4.8 \cdot 10^4 \ \mu$m. See **Figure 2—source data 1**.

The online version of this article includes the following figure supplement(s) for figure 2:

**Source data 1.** Compressed ZIP file containing all the data plotted in the panels of **Figure 2** as CSV and TXT files.

**Figure supplement 1.** Effect of crowding.

Using both the PBM and LPM models described above, we can construct traces that look similar to the data (**Figure 2B**). To mimic the data, we only record and show traces in two dimensions and added detection noise corresponding to the level reported in the experiments (**Miné-Hattab et al., 2021**). Based on these simulations, we gather the statistics of the particle motion to create a displacement histogram representing the probability distribution of the observed step sizes between two successive measurements. For this choice of parameters (see **Figure 2** caption), both of the models and the experimental data look very similar (**Figure 2C**).

In principle, we could have expected the displacement histogram of particles inside the bulk of the focus (where traces are not close to the boundary) to look markedly different between the PBM and the LPM. While the LPM should follow the prediction from classical diffusion (given by a Gaussian radial distribution, $p(|\delta\mathbf{r}|) \propto |\delta\mathbf{r}|^2 e^{-|\delta\mathbf{r}|^2/(4D\delta t)}$ for a small interval $\delta t$ in the bulk), the PBM prediction is expected to be in general non-Gaussian because of intervals during which the particle is bound and almost immobile (as the chromatin or single-stranded DNA carrying the binding sites moves very slowly), creating a peak of very small displacements. Simulations show that departure from Gaussian displacements is most pronounced when the binding and unbinding rates are slow compared to the interval $\delta t$ (**Figure 2D**, top), but is almost undetectable when they are fast (**Figure 2D**, bottom). With our parameters, the binding rate is $k_+\rho \approx 3,000$ s-1, and $k_-$ ranges from 10 to 10,000 s-1, with $\delta t = 20$ ms. For comparison, assuming weak binding to DNA, $K_d = k_+/k_- \approx 1 \ \mu$M would give $k_- \sim 40$ s-1, and assuming strong specific binding, $K_d \sim 1$ nM, implies $k_- \sim 0.04$ s-1. We stress that there is a lot of uncertainty in the values for experimentally measured rate constants, and a recent study (**Saotome et al., 2018**) found the dissociation constant $k_d$ for Rad52 in yeast to vary between two observed sites from 5.6 nM to 101 nM. **Figure 2E** shows how the detectability of non-Gaussian displacements gets worse as $k_-\delta t$ increases, and is further degraded by the presence of measurement noise.

The experimental findings of single Rad52 molecules in yeast repair foci (**Miné-Hattab et al., 2021**) suggest that the movement inside the focus are consistent with normal diffusion and its Gaussian distribution of displacements (**Figure 2C**). This observation excludes a wide range of slow binding and unbinding rates in the PBM, as this would lead to non-Gaussian statistics (**Figure 2D**, top). However, it does not rule out the PBM itself, which is undistinguishable from classical diffusion for fast binding and unbinding rates (**Figure 2D**, bottom). In addition, separating displacements inside the focus from boundary-crossing ones can be very difficult in practice, and errors in that classification may result in spurious non-Gaussian displacement distributions that would confound this test. Therefore, it is important to find observables that can distinguish the two underlying models.

## Effective description of the polymer bridging model

Motivated by experimental observations, we want to find a coarse-grained description of the PBM that can be reduced to a classical diffusion process under an effective potential and with an effective position-dependent diffusivity, and relate its parameters to the properties of the binding sites. To do so, we analyze the PBM in a mean-field approximation, which is valid in the limit where binding and unbinding events are fast relative to the traveling time of the particles. In this regime, a particle rapidly finds binding sites with rate $k_+\rho(\mathbf{r})$ (where $\rho(\mathbf{r})$ is the density of binding sites) and unbinds from them with rate $k_-$. While in principle rebinding events complicate this picture, it has been showed that the period where rebindings to the same binding site occurs can be included in the time they are bound,

and thus can be renormalized into a lower effective unbinding rate (*Kaizu et al., 2014*). Assuming that interactions between binding sites do not affect their binding to the particle of interest, the binding rate can be approximated in the presence of partially reflecting binding sites by the Smoluchowski rate (*Nadler and Stein, 1996*; *Berezhkovskii et al., 2019*) (Appendix References):

$$k_+ = \frac{4\pi D_n r_b}{1 + \frac{D_n}{r_b \kappa}}. \tag{6}$$

If the processes of diffusion, binding, and unbinding are in equilibrium, the steady state distribution of a particle can be derived using the Boltzmann distribution. The equilibrium assumption is justified by the fact that our time of observation is much smaller than the time scales of focus formation, and that the focus is of constant size during our observations. It is possible that active fluctuations are present inside the focus, but the Rad52 molecules that we are observing are not actively involved in the chemical reactions that take place over the measurement timescale. In this sense, Rad52 can be considered a passive agent, and this description is therefore an effective description of its motion inside the focus. This is supported by the fact that the Rad52 diffusion properties look constant across our observation period.

At each position $\mathbf{r}$, the unbound state is assigned weight 1, and the bound state weight $\rho(\mathbf{r})/K_d$, where $K_d = k_-/k_+$ is the dissociation constant. Then the probability distribution of the particle's position is given by:

$$p(\mathbf{r}) \propto \left(1 + \frac{\rho(\mathbf{r})}{K_d}\right) = \frac{1}{p_u(\mathbf{r})}, \tag{7}$$

where

$$p_u(\mathbf{r}) = \frac{k_-}{k_- + k_+ \rho(\mathbf{r})} \tag{8}$$

is the probability of being unbound conditioned on being at position $\mathbf{r}$.

Here, we assume that binding and unbinding is fast compared to variations of $\rho(\mathbf{r})$ experienced by the tracked particles in the measured time intervals. This assumption holds if the density of binding sites is large, which is a fundamental assumption of the PBM. In this limit, the dynamics of particles are governed by an effective diffusion coefficient, which is a weighted average between the free diffusion of tracked molecules, and the diffusion coefficient of the binding sites:

$$\tilde{D}(\mathbf{r}) = p_u(\mathbf{r})D_n + (1 - p_u(\mathbf{r}))D_b = \frac{D_n k_- + D_b k_+ \rho(\mathbf{r})}{k_- + k_+ \rho(\mathbf{r})}. \tag{9}$$

Likewise, particles are pushed by an effective confinement force: when they are bound to binding sites, they follow their motion which is confined inside of the focus. The resulting drift is given by that of the binding sites, but weighted by the probability of being bound to them:

$$\begin{aligned} \langle d\mathbf{r} \rangle &= -dt(1 - p_u(\mathbf{r}))\frac{D_b}{k_B T}\nabla U_b(\mathbf{r}) \\ &= dt\left[-\frac{\tilde{D}(\mathbf{r})}{k_B T}\nabla \tilde{U}(\mathbf{r}) + \nabla \tilde{D}(\mathbf{r})\right], \end{aligned} \tag{10}$$

where in the second line, we have rewritten the dynamics in terms of an effective potential $\tilde{U}(\mathbf{r}) = k_B T \ln(1 + k_+ \rho(\mathbf{r})/k_-)$, using $\rho(\mathbf{r}) \propto e^{-U_b(\mathbf{r})/k_B T}$. Thus, the effective dynamics may be described by the Langevin equation of the same form as the LPM (1) but with the relation between $\tilde{U}(\mathbf{r})$ and $\tilde{D}(\mathbf{r})$ constrained by their dependence on $\rho(\mathbf{r})$:

$$\tilde{U}(\mathbf{r}) = k_B T \ln\left[\frac{\tilde{D}(\mathbf{r}) - D_b}{D_n - D_b}\right], \tag{11}$$

with the convention that $\tilde{U} = 0$ far away from the focus where $\rho = 0$. As a consistency check, one can verify that the equilibrium distribution $p \propto e^{-\tilde{U}/k_B T}$ gives back Equation 7. Equation 11 reveals a fundamental relation about the dynamics of molecules inside the PBM, and is therefore an important fingerprint to test the nature of foci.

## Scaling relation between concentration and diffusivity in the PBM

Experiments or simulations give us access to the *effective diffusivity* through the maximum likelihood estimator $\tilde{D} = \langle \delta \mathbf{r}^2 \rangle / (2d\delta t)$, where $\delta t$ is the time between successive measurements, $\delta \mathbf{r}$ is the measured displacement between two measurements, and $d$ the dimension in which motion is observed. Within the PBM, Equation 11 allows us to establish a general relation between the particle concentration $p(r)$, which can also be measured, and the effective diffusivity $\tilde{D}$, through:

$$p(\mathbf{r}) \propto \frac{1}{\tilde{D}(\mathbf{r}) - D_b}. \tag{12}$$

Typically in experiments we have $D_b \ll \tilde{D} \ll D_n$, in which case this relation may be approximated by $p(\mathbf{r})\tilde{D}(\mathbf{r}) = \text{const}$.

We validated Equation 12 in simulations of the PBM. We divided the radial coordinate $r$ into small windows of $10^{-3} \mu$m and plotted the measured effective diffusion coefficient $\tilde{D}(r)$, as a function of $r$ (*Figure 2F*), as well as the density of tracked particles $p(r)$ (*Figure 2G*). $\tilde{D}(r)$ takes an approximately constant value inside the focus, defined as $D_0$ by analogy with the LPM, and is equal to $D_n$ well outside the focus where diffusion is free. Likewise the density $p(r)$ decreases from $p_{in}$ inside to $p_{out}$ outside the focus. Note that $D_0$ in the PBM is not a free parameter, but rather emerges from the mean-field description and depends on the properties of binding site. We extracted those values numerically from the simulations. *Figure 2H* shows that Equation 12 predicts well the relationship between these four numbers, for a wide range of parameter choices of the PBM (varying $\kappa$ from 1 to 400 $\mu$m/s, $k_-$ from 5 to 1500 $s^{-1}$ and $\rho$ from $23873 - 47746 \mu\text{m}^{-3}$, while keeping $D_b = 5 \cdot 10^{-3} \mu\text{m}^2/s$ and the other parameters to values given by *Table 1*). While this relation was derived in the limit of fast binding and unbinding, it still holds for the slower rates explored in our parameter range (see *Figure 2H*). However, it breaks down in the limit of strong binding, when we expect to see two populations (bound and unbound), making the effective diffusion coefficient an irrelevant quantity (see *Figure 2D*).

We can compare this prediction to estimates from the experimental tracking of single Rad52 molecules in yeast repair foci (*Miné-Hattab et al., 2021*) (see Materials and methods for details), assuming that the diffusivity of the binding sites is well approximated by that of the single-stranded DNA-bound molecule Rfa1, measured to be $D_b = 5 \cdot 10^{-3} \mu\text{m}^2/s$. This experimental point, shown as a blue cross in *Figure 2H*, substantially deviates from the PBM prediction: Rad52 particles spend much more time inside the focus than would be predicted from their diffusion coefficient based on the PBM. To agree with the data, the diffusion coefficient of binding sites would have to be increased to $D_b = 0.0314 \mu\text{m}^2/s$, which is almost an order of magnitude larger than what was found in experiments. The existence of multiple binding sites could in principle lead to an enhanced level of molecular crowding. This would in fact decrease the effective diffusion coefficient inside the focus, moving points of the PBM simulations in *Figure 2H* to the left, further away from the experimental observation. However, we checked numerically that this effect was small, by adding inert spheres of the same size as the binding sites to generate crowding (*Figure 2—figure supplement 1*).

## Diffusion coefficient and concentration predict boundary movement in the PBM

Another observable that is accessible through simulations and experiments is the radial displacement near the focus boundary. In practice, we gather experimental traces around the focus, and estimate the radius of the focus as shown in *Figure 3A*. Using many traces, we can find the average radial displacement $\langle \delta r \rangle$ during $\delta t$, as a function of the initial radial position of the particle $r$ (*Figure 3B*). Under the assumption of spherical symmetry, within the PBM this displacement is given by:

$$\begin{aligned} \langle \delta r \rangle &\simeq \delta t \left( -(1 - p_u(r)) \frac{D_b}{k_B T} \partial_r U_b(r) + \frac{\tilde{D}(r)}{r} \right) \\ &= \delta t \left( -\frac{\tilde{D}(r)}{k_B T} \partial_r \tilde{U}(r) + \partial_r \tilde{D}(r) + \frac{\tilde{D}(r)}{r} \right), \end{aligned} \tag{13}$$

where the term $\tilde{D}/r$ comes from the change to spherical coordinates.

The first line of Equation 13 shows that the average change in radial position of single particles $\langle \delta r \rangle$ cannot be negative in the PBM for steady binding sites ($D_b = 0$). This result does not hold for moving

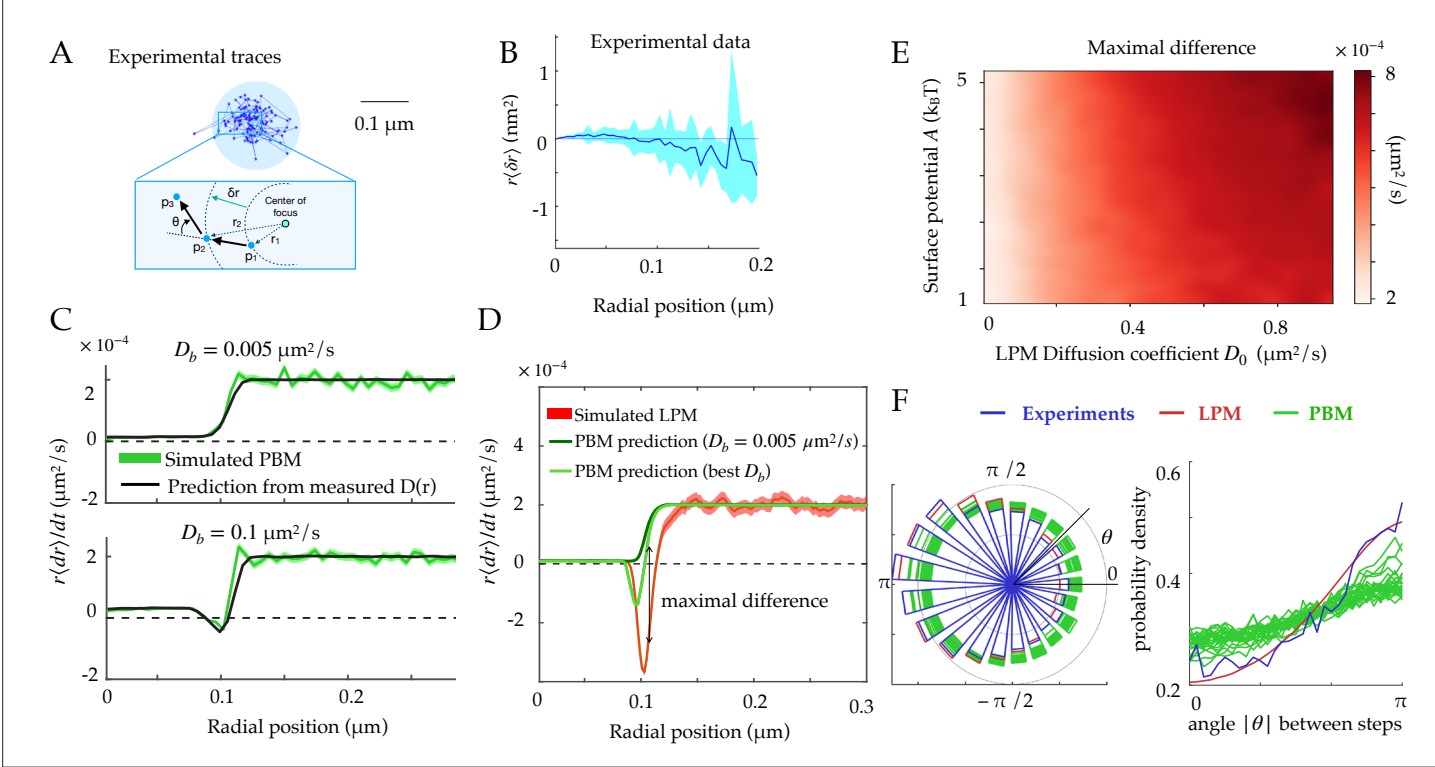

**Figure 3.** Radial and angular dynamics. (**A**) Experimental trace of a single Rad52 in a DSB focus (*Miné-Hattab et al., 2021*). The inset shows the definition of the radial movement $\delta r$. Here, concentric circles are shown to define the radius relative to the focus center, where a particle moves a specific distance away from the center of the focus, as well as the angle $\theta$ between consecutive displacements. (**B**) Data extracted from experiments (*Miné-Hattab et al., 2021*) to estimate the average radial displacement of the tracked particle multiplied by the radius. Here and throughout the x-axis represents the radial position at the beginning of the timestep. Error bars are standard errors on the mean. (**C**) Simulations showing the radial displacements in the PBM with slowly (top) and rapidly (bottom) moving binding sites. Black lines are predictions based on the measurement of $\tilde{D}$ (Equation 12). Error bars are standard deviations on the mean. (**D**) Radial displacement from simulations of the LPM. Light-green line shows the (wrong) prediction made while assuming the PBM, using the measurement of the effective diffusion coefficient $\tilde{D}(r)$ (Equation 13). We call the discrepancy between data and the PBM prediction the "Maximal positive difference." Error bars are standard errors on the mean. (**E**) Heatmap showing the maximal positive difference in LPM simulations as a function of $D_0$ and $A$. (**F**) Distribution of angles (represented radially on the left, and linearly on the right) between the displacements of consecutive steps of length $\delta t$, from experiments and simulations. Multiple curves for the PBM correspond to different parameter choices corresponding to the points of *Figure 2H*. Parameter values as in Table 1, except $r_n = 1\,\mu$ m for F. For F parameters are varied with $\kappa = 100–300\,\mu$ m/s, $k_- = 500–1,500$ s−1, $r_f = 0.1–0.14$ m. See *Figure 3—source data 1* and *Figure 3—source data 2*.

The online version of this article includes the following figure supplement(s) for figure 3:

**Source data 1.** Compressed ZIP file containing all the data plotted in the panels of *Figure 3* as CSV files.

**Source data 2.** Compressed ZIP file containing all the data plotted in *Figure 3—figure supplement 1* as CSV files.

**Figure supplement 1.** Asymmetry coefficient for an infinite PBM focus.

binding sites ($D_b > 0$), as we will see below. This is reproduced in simulations, for different absorption probabilities, as shown in *Figure 3C*.

By constrast, in the LPM there is no constraint on the sign of the displacement $\langle \delta r \rangle$ since the relation between the diffusion coefficient and the surface potential is not constrained like in the PBM. Even when binding sites can move, this prediction can be used to falsify the PBM. Equation 13 makes a prediction for the average radial displacement of the tracked molecule in the PBM, solely as a function of the diffusivity and concentration profiles $\tilde{D}(r)$ and $p(r)$, using $\tilde{U}(r) = k_B T \ln p(r)$. Accordingly, this prediction agrees well with simulations of the PBM (*Figure 3C*).

Using Equation 13 that is derived for the PBM, along with the definition of $\tilde{U}$ as a function of $\tilde{D}$ in Equation 11, to analyze a simulation of the LPM leads to large disagreement between the inferred and true parameters. This PBM-based analysis underestimates the depth of the potential (*Figure 3D*, green lines compared to the red). It predicts a negative displacement $\langle dr \rangle$ when $D_b$ is inferred using

the PBM formula $p_{\text{in}}/p_{\text{out}} = (D_n - D_b)/(D_0 - D_b)$, although its magnitude is underestimated. But when taking the experimental value of $D_b = 0.005 \ \mu\text{m}^2/s$, $\langle dr \rangle$ is always positive even at the boundary. This spurious entropic 'reflection' is an artifact of using the wrong model, since the distinct relation between the observed diffusion coefficient and the equilibrium distribution for the PBM leads to a specific shape around the boundary which is not the same for the LPM. The inference using the PBM of such a positive displacement at the surface of the focus can therefore be used to reject the PBM. *Figure 3E* represents the magnitude of that discrepancy as a function of two LPM parameters — diffusivity inside the droplet and surface potential — showing that the PBM is easier to reject when diffusivity inside the focus is high.

In summary, the average radial diffusion coefficient can predict the radial displacement of tracked molecules within the PBM, and deviations from that prediction can be used as a means to reject the PBM using single-particle tracking experiments.

## Distribution of angles between consecutive time steps

To go beyond the average radial displacement, we considered a commonly used observable to study diffusive motion in complex environment: the distribution of angles between two consecutive displacements in two dimensions. While this distribution is uniform for a homogenous environment (*Liao et al., 2012*), it is expected to be asymmetric in presence of confinement and obstacles (*Izeddin et al., 2014*).

We computed this distribution from simulations of the PBM and LPM, and compared them to experiments in yeast repair foci (*Figure 3F*), calculating the angle between the vector relating the first two points and the vector relating the last two points. These distributions are all asymmetric, with an enrichment of motion reversals (180 degree angles). Since the LPM assumes standard diffusion within a potential, the asymmetry in that model can be entirely explained by the effect of confinement, which tends to push back particles at the focus boundary. With the parameters of *Table 1*, the LPM agrees best with the data, while the PBM shows a more moderate asymmetry across a wide range of parameters. Therefore, both the LPM and the PBM are expected to show asymmetric diffusion around the boundary of the focus, but one could expect that the PBM (and not the LPM) revealed an additional asymmetry inside the bulk of the focus, due to the interactions of the tracked molecules with the binding sites, which causes reflections and hinders motion. To isolate this effect from boundary effects, we simulated the PBM in an infinite focus with a constant density of binding sites (*Figure 3—figure supplement 1*) and found that this expectation is confirmed. However, this asymmetry is seen only when the measurement time step is small or comparable to the binding time. For finite foci, it must also be corrected for boundary effects. These difficulties make the asymmetry criterion unfit to discriminate between the two models in the context of yeast repair foci.

## Foci accelerate the time to find a target, but only moderately in the PBM

Foci keep a higher concentration of molecules of interest within them through an effective potential. We wondered if this enhanced concentration of molecules could act as a 'funnel' allowing molecules to find their target (promoter for a transcription factor, repair site, etc) faster.

To address this question, we consider an idealized setting with spherical symmetry, in which the target is a small sphere of radius $r_0$ located at the center of the focus, of radius $r_f$ (*Figure 4A*). We further assume that the nucleus is a larger sphere of radius $r_n$, centered at the same position. We start from a general Langevin equation of the form in Equation 1, and assume that the target is perfectly absorbing, creating a probability flux $J = \tau_a^{-1}$, equal to the rate of finding the target for a single particle. The corresponding Fokker-Planck equation can be solved at steady state, giving (Appendix A):

$$\tau_a = \int_{r_0}^{r_n} dr \, r^2 e^{-U(r)/k_B T} \int_{r_0}^{r} \frac{dr'}{D(r')r'^2} e^{U(r')/k_B T}. \tag{14}$$

Taking the particular form of Equations 3; 4, with a sharp boundary $br_f \gg 1$, the integral can be computed explicitly:

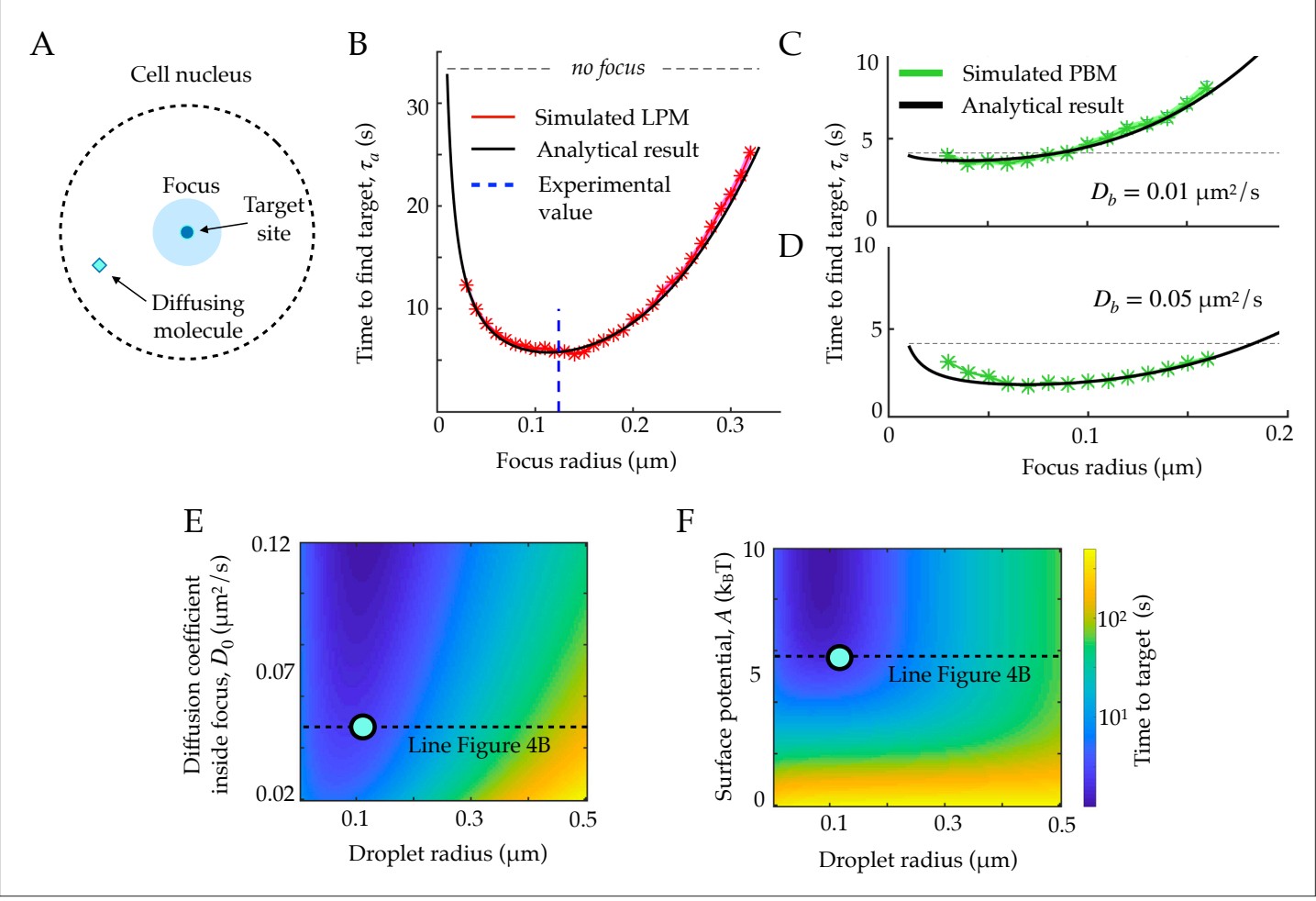

**Figure 4.** First passage times to a target site inside the focus. A. Schematic figure showing the setup of the tracked molecule and the effective target. B. Time to reach the specific target in simulations for the LPM. Black curve shows the predicted result from the analytical derivation (Equation 15). Here we use parameters: $A = -5.5k_BT$, $D_0 = 0.05\mu^2/s$, $r_n = 1.0\mu m$, $D_n = 0.8\mu^2/s$. C. Same as B, but for the PBM. Same parameters as in B, but with $D_0 = D_n \cdot e^{U(r)/k_BT}$. D. Heatmap showing the expected search time as a function of the droplet size (x-axis) and the focus diffusion coefficient (y-axis) (Equation 15). Green point corresponds to experimental observations. E. Heatmap showing the expected search time as a function of the droplet size (x-axis) and the height of the surface potential (y-axis) (Equation 15). Green point corresponds to experimental observations. Parameter values as in Table 1, except $b = 2,000\ \mu m^{-1}$, $D_0 = 0.04\ \mu m^2/s$, $A = 5.5k_BT$ for B, E, and F, $\kappa = 50\ \mu m^2/s$, $k_- = 100$ s⁻¹, $\rho = 2.4 \cdot 10^4\ \mu m^{-3}$ for C-D. See **Figure 4—source data 1**.

The online version of this article includes the following figure supplement(s) for figure 4:

**Source data 1.** Compressed ZIP file containing all the data plotted in the panels of **Figure 4** as CSV files.

$$\tau_a = \frac{r_f^3 - r_0^3}{3D_0 r_0} + \frac{r_0^2 - r_f^2}{2D_0} + e^{-\frac{A}{k_BT}}\left(\frac{r_n^3 - r_f^3}{3D_0 r_0} + \frac{r_f^3 - r_n^3}{3D_0 r_f}\right)$$
$$+ \frac{r_n^3 - r_f^3}{3D_n r_f} + \frac{r_f^2 - r_n^2}{2D_n}.$$

(15)

In the limit $r_0 \ll r_f \ll r_n$ and of a strong potential $A \gg k_BT$, Equation 15 simplifies to:

$$\tau_a \approx \frac{r_f^3}{3D_0 r_0} + \frac{r_n^3}{3D_n r_f},$$

(16)

which is exactly the sum of the time it takes to find the focus from the edge of the nucleus, and the time it takes to find the target from the focus boundary.

Expression (15) can be related to the celebrated Berg and Purcell bound (**Berg and Purcell, 1977**), which sets the limit on the accuracy of sensing small ligand concentration by a small target, due to the limited number of binding events during some time $t$. This bound puts a physical constraint on the accuracy of biochemical signaling, and has been shown to be relevant in the context of gene regulation (**Gregor et al., 2007**). With a mean concentration of ligands $c$ in the cell nucleus, there are $m = (4\pi/3)r_n^3 c$ such ligands, and their rate of arrival at the target is $m/\tau_a = 4\pi c r_n^3/(3\tau_a)$, so that the number of binding events during $t$ is equal to $n \sim 4\pi c r_n^3 t/(3\tau_a)$ on average. Random Poisson fluctuations of $n$ result in an irreducible error in the estimate of the concentration $c$:

$$\frac{\delta c^2}{c^2} \sim \frac{\delta n^2}{n^2} \sim \frac{1}{n} \sim \frac{3\tau_a}{4\pi c r_n^3 t}. \tag{17}$$

Replacing $\tau_a$ in Equation 17 with the expression in Equation 16, we obtain in the limit of large nuclei ($r_n \to \infty$):

$$\frac{\delta c^2}{c^2} \sim \frac{1}{4\pi c t}\left[\frac{1}{D_n r_f} + \frac{e^{-A/k_B T}}{D_0}\left(\frac{1}{r_0} - \frac{1}{r_f}\right)\right]. \tag{18}$$

One can further check that in the limit of a strong potential, or when there is no focus, $r_0 = r_f$, we recover the usual Berg and Purcell limit for a perfectly absorbing spherical measurement device, $\delta c/c \sim 1/\sqrt{4\pi D_n c r_f t}$.

Equation 15 agrees well with simulations in the general case (**Figure 4B**), where we used parameters obtained for Rad52 in a repair focus (**Miné-Hattab et al., 2021**). Equation 15 typically admits a minimum as a function of $r_f$, meaning that there exists an optimal focus size that minimizes the search time. Using the measured parameters for Rad52, we find an optimal focus size of $r_f^* \approx 120$ nm, which matches the estimated droplet size $r_f = 124$ nm in these experiments (**Miné-Hattab et al., 2021**) (dashed line in **Figure 4B**). In these experiments, the estimated experimental noise level was $\approx 30$ nm, but $r_f$ could be extracted accurately by fitting the confinement radius as well as gathering statistics for the radial steady state distribution. We observe that the theoretical curve in **Figure 4B** is rather flat around its minimum, suggesting an optimal range of droplet sizes rather than a single one. In the limit where $r_n \gg r_0$, the optimal size takes the explicit form:

$$(r_f^*)^4 = r_0 r_n^3 \frac{\frac{D_0}{D_n} - e^{-\frac{A}{k_B T}}}{3(1 - e^{-\frac{A}{k_B T}})} = r_0 r_n^3 \frac{\frac{D_0}{D_n} - \frac{p_{\text{out}}}{p_{\text{in}}}}{3\left(1 - \frac{p_{\text{out}}}{p_{\text{in}}}\right)}. \tag{19}$$

This optimum only exists for $D_0 e_n^{A/k_B T}$ or $D_0 p_{\text{in}} n p_{\text{out}}$, that is, when the benefit of spending more time in the focus compensates the decreased diffusion coefficient. Incidentally, in that case the Berg and Purcell bound on sensing accuracy generalizes to:

$$\frac{\delta c^2}{c^2} \sim \frac{1}{\pi c t}\left(\frac{p_{\text{out}}}{4p D_0 r_0} + \frac{1}{3D_n r_f}\right). \tag{20}$$

The previous formulas for the search time and sensing accuracy are valid for the general Langevin Equation 1, which describes both the LPM and the PBM in the mean-field regime. **Figure 4C and D** show the search time as a function of the focus size for the specific case of the PBM, where diffusion and potential are further linked. The relation between $\tilde{U}$ and $\tilde{D}$, given by Equation 11, imposes $D_0 e^{A/k_B T} = D_n + D_b(e^{A/k_B T} - 1)_n$, giving the optimal focus size:

$$(r_f^*)^4 = r_0 r_n^3 \frac{D_b}{3D_n}. \tag{21}$$

For the physiologically relevant regime of very slow binding sites, $D_b \ll D_n$, this optimal focus size shrinks to 0, meaning that the focus offers no benefit in terms of search time, because binding sites 'sequester' or 'titre out' the molecule, preventing it from reaching its true target.

These results suggest to use the search time, or equivalently the rate for binding to a specific target, as another measure to discriminate between the LPM and the PBM. In the case of slowly diffusing binding sites, the search time in the PBM does not have a clear local minimum (see **Figure 4D**), and depends less sharply on the focus size than in the LPM. Therefore, identifying an optimal focus size would suggest to rule out the PBM. Conversely, a monotonic relation between the search time and the focus size would be consistent with the PBM (without excluding the LPM). Testing for the existence

of such a minimum would require experiments where the focus size may vary, and where reaching the target can be related to a measurable quantity, such as gene expression onset in the context of gene regulation.

## Discussion

The PBM and the LPM are the two leading physical models for describing the nature of nuclear foci or sub-compartments. In this work, we analyzed how the traces of single particle tracking experiments should behave in both models. Using statistical mechanics, we derived a mean field description of the PBM that shares the general functional form of the LPM (Equation 1), but with an additional constraint linking concentration and diffusion inside the focus: the denser the focus, the higher the viscosity. This constraint does not appear to be satisfied by the experimental data on Rad52 in repair foci, favoring the liquid droplet hypothesis. We use our formulation of the PBM to predict the behaviour of the mean radial movement around the focus boundary, which may differ markedly from observation of traces inside a liquid droplet (described by the LPM). We find the range of LPM parameters where this difference would be so significant that it would lead to ruling out the PBM. This work provides a framework for distinguishing the LPM and PBM, and should be combined with modern inference techniques to accurately account for experimental noise and limited data availability (for instance accounting for molecules going out of the optimal focus). Future improvements in single-particle tracking experiments will allow for longer and more accurate traces necessary to deploy the full potential of these methods.

The LPM and PBM have often been presented as opposing models (*Miné-Hattab and Taddei, 2019*), driven by attempts to compare the macroscopic properties of different membraneless sub-compartments to the original example of liquid-like P granules (*Brangwynne et al., 2009*). The LPM is a macroscopic description of a liquid droplet in the cytoplasm (*Hyman et al., 2014*), which concentrates some molecules inside the droplet, and alters their different diffusion properties. The droplet is formed by a phase transition, which means it will be recreated if destroyed, and will go back to its spherical shape if sheared or merged. Conversely, the PBM describes the motion and effective diffusion coefficient inside the focus as a result of fundamental interactions, which provides an explicit binding mechanism by which a focus is formed. Here, we clarified the link between the two from the point of view of single molecules. We confirmed mathematically the intuition that, in the limit of very fast binding and unbinding, the PBM is a particular case of the LPM model. Going further, we show that the PBM imposes a strong constraint between the effective diffusion of molecules in the subcompartment, $D(\mathbf{r})$, and the effective potential, $\tilde{U}(\mathbf{r})$ (Equation 11). The LPM is compatible with this choice, but does not impose it in general, although alternative mechanistic implementations of the LPM may impose similar constraints with different functional forms. The correspondence between the two models breaks down when binding and unbinding are slow. However, for this regime to be relevant, experimental observations need to be fast enough to capture individual binding or unbinding events, which is expected to be hard in general, and was not observed in the case of repair foci in yeast.

We found another way in which the two models behave very differently: in the LPM, the focus may act as a funnel accelerating the search for a target inside the focus, and we calculated the optimal focus size that minimizes the search time. In the PBM, such an improvement is negligible unless binding sites themselves have a fast diffusive motion. This difference between the two models could potentially be tested in experiments where the focus size varies. It is not clear whether this optimality argument is relevant for DSB: the merger of two foci leads to larger condensates, suggesting that the focus size is not tightly controlled. But the argument may be relevant for gene expression foci, especially in the context of development where transcription factors need to reach their regulatory target fast in order to ensure rapid cell-fate decision making (*Bialek et al., 2019*). On the contrary, if a focus is created in order to decrease the probability of specific binding, such as in silencing foci (*Brown et al., 1997*), a PBM implementation may be more advantageous. Binding sites, which act as decoys (*Burger et al., 2010*), sequester proteins involved in gene activation, thus increasing the time it takes to reach their target and suppressing gene expression. In that picture, genes would be regulated by the mobility and condensation of these decoy binding sites. Therefore, while this difference between the two models may be hard to investigate experimentally, it provides be a very important distinction in terms of function.

More generally, foci or membraneless sub-compartments are formed in the cells for very different reasons and remain stable for different timescales. For example, repair foci are formed for short periods of time (hours) to repair double strand breaks, and then dissolve. In this case the speeds of both focus formation and target finding are important for rapid repair, but long-term stability of foci is not needed. Gene expression foci (*Hnisz et al., 2017*; *Bing et al., 2020*) can be long lived, and their formation may be viewed as a way to 'prime' genes for faster activation. However, given the high concentrations of certain activators, not all genes may require very fast search times of the transcription factors to the promoter. While molecularly the same basic elements are available for foci formation – binding and diffusion – different parameter regimes exploited in the LPM and PBM may lead to different behaviour covering a vast range of distinct biological requirements.

## Materials and methods
### Simulation of PBM

In order to simulate the bridging model we generated $N$ binding sites of radius $r_b$. We simulate a diffusing molecule through the free overdamped Langevin equation in three dimensions, and at each time-step, we find the closest binding site to the particle. If the distance of the particle ($\Delta r$) is smaller than $r_b$ we bind the molecule with probability $p_b = \kappa\sqrt{\pi\delta t/D_n}$. If the particle does not bind, it is reflected so the new distance to the center of the particular binding site is $2r_b - \Delta r$. At this new position we evaluate the position of all other binding sites they all diffuse with diffusion coefficient $D_b$, and if the molecule is within the radius of another binding site (happens extremely rarely), it is again accepted to bind with the same probability $p_b$. If a particle binds, it stays at the position of the intersection with the binding site, and at each time step it can be released with probability $k_{-}\delta t$. We choose $\delta t$ small so that $p_b \ll 1$ and $\sqrt{2D_n\delta t} \ll r_b$, which for the considered parameter ranges in *Table 1* is typically obtained for values of $\delta t = 10^{-6}s$.

### Simulation of LPM

To simulate the LPM, we use the Milstein algorithm to calculate the motion of a particle. As in the PBM, the particle is reflected at the nucleus boundary, and can otherwise move freely in the nucleus. We typically choose the same value of $\delta t$ as the PBM, since the surface potential typically has a very steep gradient, given by $b \approx 1000$ as shown in *Table 1*.

### Experimental measurements

Experimental details about single-particle tracking are given in *Miné-Hattab et al., 2021*. Briefly, the x- and y-values of single particles were sampled at 50 Hz for molecules inside the visible z-frame ($\approx \pm 150nm$ thick). Therefore one cannot separate whether molecules are inside the focus or above/below it, but since the radius of the focus is $\approx 125nm$, this effect is very small, and statistically it is possible to take this effect into account when calculating the radial concentration of molecules.

The diffusion coefficient inside the focus was calculated as follows. The distributions of displacements was fitted by a mixture of two Gaussians corresponding to a slow (inside focus) and a fast (outside focus) population. Diffusion inside the focus was extracted from the mean-squared displacement of the slow population, taking the confinement and experimental uncertainty into account (see text related to Figure 2G in *Miné-Hattab et al., 2021*). Free energy differences were estimated based on the size of the focus and the concentration of particles inside the focus compared to outside (see text related to Figure 7 and Table S1, *ibid.*). These estimates are not sensitive to radial effects, such as the definition and size of the focus, or to the issue of some particles being above or below the focus.

## Acknowledgements

The authors are grateful to Jean-Baptiste Masson, Alexander Serov, Namiko Mitarai and Ned Wingreen for valuable discussions. The study was supported by the Agence Nationale de la Recherche (Q-life ANR-17-CONV-0005), Centre National de la Recherche Scientifique (80' MITI project PhONeS), the European Research Council COG 724208, the Labex DEEP (ANR-11-LABEX-0044 DEEP and ANR-10-IDEX-0001?02 PSL), the ANR DNA-Life (ANR-15-CE12-0007), the Fondation pour la Recherche Médicale (DEP20151234398), and the ANR-12-PDOC- 0035?01. The authors greatly acknowledge the

PICT-IBiSA@Pasteur Imaging Facility of the Institut Curie, member of the France Bioimaging National Infrastructure (ANR-10-INBS-04). MLH acknowledges the Carlsberg Foundation grant CF20-0621.

## Additional information

### Competing interests

Aleksandra M Walczak: eLife senior editor. The other authors declare that no competing interests exist.

### Funding

| Funder | Grant reference number | Author |
|---|---|---|
| Agence Nationale de la Recherche | Q-life 356 ANR-17-CONV-0005 | Mathias L Heltberg Judith Miné-Hattab Angela Taddei Aleksandra M Walczak Thierry Mora |
| Centre National de la Recherche Scientifique | 80' MITI project PhONeS | Judith Miné-Hattab Angela Taddei |
| H2020 European Research Council | COG 724208 | Mathias L Heltberg Aleksandra M Walczak Thierry Mora |
| Agence Nationale de la Recherche | ANR-15-CE12-0007 | Judith Miné-Hattab Angela Taddei |
| Agence Nationale de la Recherche | ANR-12-PDOC- 0035?01 | Judith Miné-Hattab Angela Taddei |
| Fondation pour la Recherche Médicale | DEP20151234398 | Judith Miné-Hattab Angela Taddei |

The funders had no role in study design, data collection and interpretation, or the decision to submit the work for publication.

### Author contributions

Mathias L Heltberg, Conceptualization, Investigation, Methodology, Software, Visualization, Writing - original draft, Writing - review and editing; Judith Miné-Hattab, Angela Taddei, Conceptualization, Funding acquisition, Resources, Writing - review and editing; Aleksandra M Walczak, Thierry Mora, Conceptualization, Funding acquisition, Investigation, Methodology, Supervision, Writing - original draft, Writing - review and editing

### Author ORCIDs

Mathias L Heltberg ⬤ http://orcid.org/0000-0002-9699-4075
Judith Miné-Hattab ⬤ http://orcid.org/0000-0001-9986-4092
Angela Taddei ⬤ http://orcid.org/0000-0002-3217-0739
Aleksandra M Walczak ⬤ http://orcid.org/0000-0002-2686-5702
Thierry Mora ⬤ http://orcid.org/0000-0002-5456-9361

### Decision letter and Author response

Decision letter https://doi.org/10.7554/eLife.69181.sa1
Author response https://doi.org/10.7554/eLife.69181.sa2

## Additional files

### Supplementary files

• Transparent reporting form

### Data availability

All data generated or analysed during this study are included in the manuscript and supporting files.

The following previously published datasets were used:

| Author(s) | Year | Dataset title | Dataset URL | Database and Identifier |
|---|---|---|---|---|
| Miné-Hattab J, Heltberg M, Villemeur M, Guedj C, Mora T, Walczak AM, Dahan M, Taddei A | 2021 | Single molecule microscopy reveals key physical features of repair foci in living cells | https://zenodo.org/record/4495116 | Zenodo, 4495116 |

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

## Appendix 1

### Binding rate by a partially absorbing sphere

We consider a particle with diffusivity $D$, which can be partially absorbed by a spherical binding site of radius $r_b$ and absorption parameter $\kappa$. Its Fokker-Planck equation takes the following form, in spherical coordinates projected onto the distance to the center of the binding site, $r$:

$$\partial_t p = \frac{D}{r^2} \partial_r r^2 \partial_r p. \tag{22}$$

The boundary conditions are $p(r = \infty) = 1/V$, where $V$ is the total volume, assumed to be much larger than that of the binding site, and the Robin condition:

$$D \partial_r p(r_b) = \kappa p(r_b). \tag{23}$$

The solution of Equation 22 at steady state with these bondary conditions reads:

$$p(r) = \frac{1}{V} \left( 1 - \frac{\kappa r_b / r}{\kappa + D/r_b} \right), \tag{24}$$

the total diffusive flux is then given by

$$J = 4\pi D r_b^2 \partial_r p(r_b) = \frac{1}{V} \frac{4\pi D r_b}{1 + \frac{D}{r_b \kappa}}. \tag{25}$$

Normalizing by the volume factor gives the association rate for binding, $k_+ = VJ = 4\pi D r_b / (1 + D/\kappa r_b)$.

## Appendix 2

### Searching for a target in a funneling potential

We consider a problem similar to that of the previous appendix. Now the spherical object is a target, which is perfectly absorbing. It is at the center of a liquid droplet, which we model by a spherically symmetric potential $U(\mathbf{r})$.

The probability distribution of a molecule is denoted by $p(\mathbf{r}) = p(r)$. The probability density of being at distance $r$ from the center, $q(r)$, is related to $p(r)$ through $q(r) = 4\pi r^2 p(r)$, accounting for the volume of the sphere. The evolution of $r$ is described by the stochastic differential equation:

$$dr = \frac{2D}{r} + \partial_r D - \frac{D}{k_B T}\partial_r U + \sqrt{2D}dW, \tag{26}$$

where $W$ is a 1-dimensional Wiener process. The corresponding Fokker-Planck equation reads:

$$\partial_t q = -\partial_r \left[\left(\frac{2D}{r} + \partial_r D - \frac{D}{k_B T}\partial_r U\right)q\right] + \partial_r^2(Dq) \doteq -\partial_r J. \tag{27}$$

At steady state with a non-vanishing flux $J = \text{const}$, we have:

$$\left(\frac{2D}{r} - \frac{D}{k_B T}\partial_r U\right)q = D\partial_r q - J, \tag{28}$$

or equivalently:

$$q\partial_r \phi + \partial_r q = \frac{J}{D} \tag{29}$$

with $\phi \doteq -2\ln(r) + U/k_B T$. Multiplying both sides of the equation by $e^\phi$, we obtain:

$$\partial_r(e^\phi q) = \frac{J}{D}e^\phi. \tag{30}$$

The general solution to that equation is:

$$q(r) = Ce^{-\phi(r)} + Je^{-\phi(r)}\int_{r_0}^r \frac{e^{\phi(r')}}{D(r')}dr'. \tag{31}$$

We have $C = 0$ because of the absorbing boundary condition $q(r_0) = 0$. The constant $J$ is determined by the normalization $\int_{r_0}^{r_n} dr\, q(r) = 1$, yielding:

$$J^{-1} \doteq \tau_a = \int_{r_0}^{r_n} dr\, e^{-\phi(r)}\int_{r_0}^r dr'\frac{e^{\phi(r')}}{D(r')}, \tag{32}$$

This in turns gives the result of the main text after replacing $\phi(r)$ by its definition.

