## [Editor Report]

There has been a lively debate recently concerning the multiplicity of reported observations of phase-separated compartments inside of cells. Specifically, some claims of phase separation have been challenged, and an alternative model has been put forward that explains clustering of observed particles as resulting from colocalization of binding sites with no phase separation. The current study does an admirable job of proposing and analyzing ways of distinguishing these two scenarios.

---

## [Decision Letter]

**Decision letter after peer review:**

Thank you for submitting your article "Physical observables to determine the nature of membrane-less cellular sub-compartments" for consideration by *eLife*. Your article has been reviewed by 3 peer reviewers; this evaluation has been overseen by Agnese Seminara as Reviewing Editor and José Faraldo-Gómez as the Senior Editor. The following individuals involved in review of your submission have agreed to reveal their identity: Pierre Ronceray (Reviewer #3).

Essential revisions:

The reviewers believe the work is suitable for *eLife* but also believe the manuscript could be improved upon – as noted below. Please consider their concerns and recommendations and implement the suggested changes whenever possible.

*Reviewer #1:*

The authors propose several ways of leveraging single-particle tracking experiments to distinguish between intracellular phase separation and an alternative model of clustered binding sites. The first proposed scheme is particularly intuitively appealing: in the binding site scenario, the local density of binding sites both increases particle density and slows effective particle diffusion, leading to a definite relationship between these two quantities, while the phase separation scenario would not necessarily couple these two quantities. The additional schemes based on particle movement near a cluster boundary, angles between consecutive steps, and search times add to the arsenal of potential analysis tools. Overall, the work is timely, rigorous, and generally clearly presented and given the growing list of reported observations of phase separation, will appeal to a broad audience.

1. The authors don't explicitly address the effects of crowding that might occur inside a cluster of binding sites. Crowding can change both the free density and the free diffusion coefficient of the particles in the cluster. How would such crowding affect the relation between observed particle density and observed diffusion, particularly if crowding scales with density of binding sites?

2. I found the discussion of the angles between consecutive steps hard to follow at points. In particular, what do the authors have in mind by the statement that binding sites can "reflect" the motion of tracked molecules (line 219)? I also wasn't sure what the final sentence of that section was meant to convey – some more guidance on the conditions under which this approach is useful would help.

3. The paper is well written but could use some additional proofreading for spelling, e.g. "dropblet", "membranelss", "rapide", "displacememnt", "mimick".

*Reviewer #2:*

Heltberg et al., investigate two possible mechanisms for the formation of nuclear foci and how these mechanisms can be distinguished experimentally, based on single-particle tracking of molecules that are up-concentrated in the focus. First, liquid-liquid phase separation (here: Liquid Phase Model, LPM) is treated as one of the major mechanisms currently hypothesized. Second, as an alternative mechanism, a polymer-bridging model (PBM) is investigated, in which the focus is held together by polymer bridges and contains binding sites, which can lead to local enrichment, appearing as a focus.

The theory is presented in a clean way, and while the Langevin equation for single molecules in a phase-separated liquid comes without derivation, it is plausible, and in fact backed up by our own calculations. A similar Langevin equation is found for the PBM and it is subsequently shown that both models can lead to very similar displacement distributions, thus showing that this simple observable cannot always distinguish between PBM and LPM.

Subsequently, the authors derive an effective description of the PBM, based on the experimental observation that potential binding sites on the DNA (proxied by Rfa1, a DNA-binding protein) diffuse much more slowly than a typical repair factor (represented by Rad52). Thus there is a separation of time scales between the two relevant diffusion processes, which is used to constrain the possible parameter combinations for the PBM. Based on these constraints, the authors shown that PBM is incompatible with their previous experimental results.

The remainder of the paper deals with a number of interesting observables, such as the angular distribution of displacements and search time to find a repair target, which can also be used to distinguish PBM and LPM with an ideal setup.

Strengths:

Heltberg et al., present a clean way to distinguish LPM on the one hand, and a realization of PBM on the other hand, based on theory. This is validated by comparison to data they obtained in previous work. The theory is rigorous and the data analysis is well carried out, save for minor ambiguities, which can likely be eliminated during revision. The paper draws its main strength from its interdisciplinarity.

Conclusions and Discussion:

The authors have achieved their goal of distinguishing LPM and PBM. The corresponding theory will be of great use for everyone in the field aiming to make this distinction based on single molecule tracking, a strategy that has been attempted numerous times, but eventually always failed due to the lack of an appropriate theoretical framework. Heltberg et al., have gone on to show a striking difference between experimentally constrained PBM realizations and the experimental measurements themselves, rendering the PBM much less likely than the LPM.

The manuscript could use some additional proofreading for grammar/typos.

Regarding the references to Miné-Hattab et al., 2021 we were uncertain with regard to two points:

1. Figure 4b shows a striking agreement between the focus size and the minimum in the time-to-find-a-target function. Measuring the focus radius of 120 nm with such a high accuracy requires exquisite microscope resolution, while we believe that this is in principle possible with PALM, we couldn't find any supporting data for this resolution in the original paper, would it be possible to point us either to the right place in the manuscript or to general references that make clear that this is a standard resolution for live-PALM the way it was used in Miné-Hattab et al.?

2. Please point us to the measurements of inverse partitioning (p_out_/p_in_). We weren't able to immediately find them in the previous paper.

Abstract/Intro:

Foci as a word works, however, in particular in the intro condensate and foci need more of distinction. A nucleolus (as a very prominent example of a nuclear condensate) wouldn't be called focus in the literature.

The abstract seems to be missing a key point of the paper, namely that based on the available observables/data a call can be made that yeast repair foci are more likely LPM, rather than PBM.

L48: The references seem to be a mix of reviews and primary research, we believe this strong statement calls for only primary references.

L54: What did Miné-Hattab et al., find? How did you make the call in your previous paper?

Results:

L73: maybe 'by the Boltzmann factor' is a bit easier to google for the uninitiated reader, rather than 'Boltzmann's law', which isn't really a thing.

L76: 'While this description is general' is unclear, but could simply be left out.

L80ff: Motivate why this potential is a good way to model binding sites? In the PMB model, the diffusion model for the binding sites ends up being very similar to the one used for LPM. Is it then fair to claim that this model is more "microscopic"? It seems that PBM is more microscopic only in words and not really in the modeling. One might say that the binding sites are just themselves following diffusion in an LPM. Then the difference between the two models is more about the fact that for LPM the tracer molecule diffuses in the LPM and in the other case the tracers do not feel directly the LPM but just bind to molecules (or binding sites) that are diffusing in the LPM. In this respect, the distinction between the two models does not contain much information on whether the foci are formed by phase separation or not. In view of the importance of this distinction and the ongoing controversies, the authors should clearly discuss this issue. To have a more microscopic model it might be useful to adopt a polymer model description, rather than a simple potential.

L92: Why exclude potential? Relevant for confinement, shape of droplet boundary (which might be a significant part of the droplet, for such small droplets). Also, six isn't much worse than five parameters.

L105: Maybe say that parameter values are mentioned in Figure legend.

L109: Why just the simple Gaussian of free diffusion? Shouldn't there be confinement effects for small droplets?

L116: Would be good to have references for k- in yeast. At least ballpark. Otherwise also look at other k-.

L120: Does Rad52 ever appear in groups, e.g. dimers or multiple times within the same repair site?

L127ff: This entire section is a bit less clear than the rest. Maybe a slightly longer, more systematic exposition could help?

L131: Rebinding description is a bit vague, maybe elaborate a bit? Can this be connected to appendix 1?

Equation 7: Shouldn't the second ∝ be an equal sign?

141: How realistic is this limit? Doesn't it imply also fast diffusion? In other words, it seems like equation 6 gives many constraints on what your rate can be. Please comment. How well does the time scale separation hold? If it doesn't, many of the conclusions would have to be modified.

L152: Recall the definition of δ r. Is average from equation 10 removed?

L161: Not sure about D_0_, according to table this is only part of LPM? Also not mentioned in L93.

L164: Typo in unit for rho.

L167: Is this shown? Is this Figure 2d?

L181 and equation 13: Inconsistent use of δ vs d.

Equation 13: In measuring this in 3C, is the average r as a function of r computed from displacements starting at r or arriving at r?

L164: Is it obvious that dr cannot be negative? What are the consequences of this? Is it true at the focus boundary?

L195: Why underestimate and how is this seen? Underestimate with respect to what ground truth? Overall the discussion of 3C, D should be extended and made clearer. It is one of the main signatures used to tell PBM from LPM so it should be very clearly discussed.

L197: Didn't you say above it was never negative? Now only for this value?

L199: Can you expand this a bit for a clearer picture?

L210: Without confinement in shouldn't it be uniform in any dimension?

L250: Can you comment more on the Burger Purcell limit? How relevant is the discussion in the present context? If it is relevant, it should be expanded.

L263: Missing 'of'.

L269ff: This seems quite weak as a test, compared to the previous results (in particular 2H). The PBM clearly also has a minimum, how different are these for different parameter ranges? It also seems very hard to control focus size in vivo.

L297: Using 'condensates' as a verb required rereading this sentence for clarification, maybe rephrase.

Figures:

General: It would be helpful to have bold one-sentence figure caption summaries. Please make sure to always mention the equations that each panel refers to in the figure caption.

1: Panel B: might be good to also have shading for the background of the nucleus. Otherwise the boundary could be mistaken for a nuclear membrane happened to us at first glance (also compare Figure 4b).

2: Panel D: caption: are (bottom) and (top) reversed?

Panel F: Caption: please clarify 'using displacement histograms'.

Panel G: Displacement is misleading for the x-axis label, how about 'r' or something like 'dist. from nucleus'.

Panel H: reorder legend, otherwise data point 'experimental observation' looks like legend.

X axis has D_0_, shouldn't this be D^~^? Same in caption for H.

To us this panel has the strongest evidence that the model of choice should be LPM,

is this true? If yes, please clarify in text.

Caption for H: Could you refer to the figures/sections in Miné-Hattab where the

partitioning is calculated? After a quick skim we weren't able to find it.

3: Panel B: What do the error bars refer to?

Panel C: Why the dip in the bottom panel? Equation 13 is said to never be negative? Is only D_b_ varied between the bottom and top panels? If D^~^ is kept constant, shouldn't the plateaus be the same?

Panel D: Caption: black instead of blue line?

Are these plots purely based on equation (13)? Same as in C, how can this be negative?

*Reviewer #3:*

Membraneless condensates have recently become a central focus of the molecular and cellular biophysics communities. While the dominant paradigm for their formation, liquid-liquid phase separation (LLPS), has been well established in a number of cases for large, optically resolved droplets, there are significant concerns regarding the generality of this mechanism for smaller foci or puncta, and other mechanisms have been proposed to explain their formation. The problem is that it is very difficult to distinguish experimentally between these mechanisms for sub-optical resolution condensates. In this article, Heltberg et al., propose a novel method, based on the analysis of single molecule tracks, that allows discriminating between the liquid phase model (LPM) and one of the challenger mechanisms, the "polymer bridging model" (PBM). This method relies on the statistics of individual displacements – diffusion, radial displacements, angular changes – which are showed theoretically to exhibit different signatures for the two models. With realistic data this is sufficient to discriminate between the models: for instance in the case of double strand break foci (DSB), building on a recent work by some of the same authors, this article convincingly rules out the PBM in favor of the LPM. The author also investigate the influence on these two models on the search time to reach a specific small target – a commonly invoked role of condensates – and show that only the LPM substantially accelerates this, which could provide additional means to experimentally discriminate between the mechanisms, on top of the intrinsic interest of this finding.

This article is a welcome addition to the literature in this field, as it will help clarify the nature of these condensates, in particular below the optical resolution. It is well-written, interesting and the conclusions are justified. I particularly appreciate the effort to employ simulated data that are realistic for actual experiments, which strengthens the claims of applicability. Some aspects of the data analysis and of the modeling, however, are insufficiently discussed and would need to be precised / expanded.

1) The modeling is made under the assumption of thermal equilibrium, without further discussion. The authors should comment on why this is reasonable, in particular in view of the presence of active fluctuations and of chemical reactions in these condensates.

2) How is the diffusivity measured? Are these measures corrected for experimental error (e.g. using three-point estimators)?

3) The conditioning of the averages should be discussed, e.g. in Equation 13: I assume that it is in the Ito convention? Similarly for the angle changes.

---

## [Author Response]

Essential revisions:The reviewers believe the work is suitable for eLife but also believe the manuscript could be improved upon – as noted below. Please consider their concerns and recommendations and implement the suggested changes whenever possible.Reviewer #1:The authors propose several ways of leveraging single-particle tracking experiments to distinguish between intracellular phase separation and an alternative model of clustered binding sites. The first proposed scheme is particularly intuitively appealing: in the binding site scenario, the local density of binding sites both increases particle density and slows effective particle diffusion, leading to a definite relationship between these two quantities, while the phase separation scenario would not necessarily couple these two quantities. The additional schemes based on particle movement near a cluster boundary, angles between consecutive steps, and search times add to the arsenal of potential analysis tools. Overall, the work is timely, rigorous, and generally clearly presented and given the growing list of reported observations of phase separation, will appeal to a broad audience.

We thank the referee for the positive attitude towards our work and are happy for the insightful comments that have increased the value of this paper. We have taken all comments into account in the point-to-point response as can be seen below.

1. The authors don't explicitly address the effects of crowding that might occur inside a cluster of binding sites. Crowding can change both the free density and the free diffusion coefficient of the particles in the cluster. How would such crowding affect the relation between observed particle density and observed diffusion, particularly if crowding scales with density of binding sites?

We agree that crowding could be an important effect of the cellular environment that should be addressed more explicitly. In the typical view of crowding, the presence of multiple large proteins affects the diffusion coefficient in the way:DpureDcrowd=ncrowdnpure

where D is the relative diffusion coefficients and η is the viscosity. Therefore crowding should affect the diffusion coefficient by making it lower, without affecting the free energy, since the interactions are assumed to be non-binding with molecules leading to crowding. This means that the equilibrium distribution of density of Rad52 particles, p_in_ and p_out_, should not be affected. Taken together if crowding should have a significant effect, this would lead to points to the upper left of the line of Figure 2H. Thereby one could still exclude the two models for a large range of parameters and this would not affect the main conclusions of this paper.

We tested this effect numerically, by replacing a fraction of the binding sites by inert reflecting spheres. We tried in the range of 20%-80%, with parameters otherwise similar to the ones used in Figure 2H. This mimics crowding since the observed particle would have more reflection events but fewer binding events. However, the effect of these reflections are minimal compared to the effect of bindings. As can be seen in figure 1 of this response, the observed points still fall on the line, which is explained by the fact that binding events are much more dominant for the observed dynamics compared to the reflection events.

Furthermore, for the results in Figure 4, the presence of large molecular crowing would increase the search time even more for the PBM, making it even worse at enhancing the on-rates of specific reactions.

We have added a few sentences in the section following the introduction of Figure 2H, in the final part of the section: ”Scaling relation…”, as well as a new supplemental figure to Figure 2.

2. I found the discussion of the angles between consecutive steps hard to follow at points. In particular, what do the authors have in mind by the statement that binding sites can "reflect" the motion of tracked molecules (line 219)? I also wasn't sure what the final sentence of that section was meant to convey – some more guidance on the conditions under which this approach is useful would help.

We have tried to add some explanatory sentences. When we use the word “reflect,” we mean that the density of binding sites is high, and since they are partially reflecting, the reflections result in an enrichment of the change of angle around π. This effect should only be seen if the time scale of measurement is small or comparable to that of binding and unbinding. On the other hand, since the focus is relatively small, a significant part of the particles hit the focus boundary, affecting the statistics even at relatively long timescales. Therefore, both the reflections from the individual binding sites (only present in the PBM) and the effective potential around the boundary of the focus (present in both the PBM and LPM) can in general lead to an asymmetry in the distributions of angles.

In the experiments, the measurement time scale is too large to see the effect of reflections against the binding sites from the PBM. However, asymetry is observed, which can be attributed to the influence of the focus boundary.

To explore the effect of reflections against binding sites, we recourse to simulations in an infinite medium (where boundaries do not play a role) with shorter observational time scales, as reported in Figure 3—figure supplement 1.

We have rewritten parts of this section in order to make this point clearer in the revised version of the manuscript.

3. The paper is well written but could use some additional proofreading for spelling, e.g. "dropblet", "membranelss", "rapide", "displacememnt", "mimick".

We thank the referee for this comment and apologize for the inconvenience. A serious proofreading has been performed for the revised version.

Reviewer #2:General:The manuscript could use some additional proofreading for grammar/typos.

We thank the referee for this comment and apologize for the inconvenience. A serious proofreading has been performed for the revised version.

Regarding the references to Miné-Hattab et al., 2021 we were uncertain with regard to two points:1. Figure 4b shows a striking agreement between the focus size and the minimum in the time-to-find-a-target function. Measuring the focus radius of 120 nm with such a high accuracy requires exquisite microscope resolution, while we believe that this is in principle possible with PALM, we couldn't find any supporting data for this resolution in the original paper, would it be possible to point us either to the right place in the manuscript or to general references that make clear that this is a standard resolution for live-PALM the way it was used in Miné-Hattab et al.?

The resolution in PALM (in fixed cells) depends on the signal to noise ratio. For our types of experiments, we obtain a standard resolution of 30 nm. For live PALM, the signal to noise ratio is smaller than in fixed cells (the fixation altering a little bit the fluorescence). We measured a pointing accuracy of 20 nm in live PALM. However, the limitation in the resolution comes from the motion of the structure during the time of the acquisition. Therefore, to optimize the acquisition, we used the shortest possible exposure time (20 ms) and performed videos of 1000 frames (20 sec). Since a focus can move in 20 seconds, we also discarded foci that moved too much in x and y during the acquisition. Finally, we also measured the mobility of the whole foci formed by 1 versus 2 DSBs, and observed similar mobilities (Figure 6B from Min´e-Hattab et al., 2021).

To support the conclusions obtained with PALM, we also derived the radius of the focus based on the MSD curves, where we separated the traces belonging to the slow diffusion coefficient. In this way we could do a fit to the MSD for confined diffusion, and extracted the best value and the uncertainty on the parameters. Here we found the same values for the radius of the foci, which confirmed that both methods measured the radius of the focus to be ≈ 120nm.

To clarify this point in the manuscript, we have added some sentences related to Figure 4B. Here we explain how the number of 124 nm was extracted based on the fit to the confinement radius. Furthermore we relax the claim that this correspondence between the number is surprising, by stressing that the theoretical curve is flat around the minimum, so there really is a range of optimal droplet radii.

2. Please point us to the measurements of inverse partitioning (p_out_/p_in_). We weren't able to immediately find them in the previous paper.

We derived the free energy of this, from which the inverse partitioning can be directly obtained. This was done in the final part of the results of Min´e-Hattab et al., 2021, related to Figure 7.

We have added a subsection of Methods to explain how various measurements were obtained in Min´e-Hattab et al., 2021.

Abstract/Intro:Foci as a word works, however, in particular in the intro condensate and foci need more of distinction. A nucleolus (as a very prominent example of a nuclear condensate) wouldn't be called focus in the literature.

We agree that the nucleolus would not be considered a typical focus, but at the current stage of the literature, a lot of terms are used not always in a clear way. This is not surprising since the field is still quite young and new results affect how we think of these structures. As argued in the paper by Banani, Lee, Hyman and Rosen from 2017 the ”non-Membrane bound compartments … have been referred to by a variety of names, including cellular bodies, nuclear bodies, membraneless organelles, granules, speckles, aggregates, assemblages, membrane puncta” and they propose the name Biomolecular condensates. Overall, we feel that ”Condensate” is more general and does not refer to a small spherical object, which foci does. Here, we consider a focus to be a spherical condensate of size smaller than a few hundreds nanometers.

We have added text to the first paragraph of the introduction to better discuss this definition.

The abstract seems to be missing a key point of the paper, namely that based on the available observables/data a call can be made that yeast repair foci are more likely LPM, rather than PBM.

We are thankful for this comment, and we agree that this is very important to mention in a clear way.

We have added a sentence to the abstract.

L48: The references seem to be a mix of reviews and primary research, we believe this strong statement calls for only primary references.

We thank the referee for this comment, and we agree on this statement. We have removed one reference, so it is now only primary references.

L54: What did Miné-Hattab et al., find? How did you make the call in your previous paper?

In the paper of “Single molecule microscopy reveals key physical features of repair foci in living cells” of Min´e-Hattab et al., 2021, we suggested (but did not conclude) that Rad52 moved according to the LPM model, which is not the case for the single binding protein Rfa1, also contained inside repair foci. This suggestion was based on 6 different observations that we will briefly list here:

– Rad52 molecules inside the focus were significantly more mobile than the focus itself and than double-strand break flanking region.

– We observed a sharp change in the diffusion coefficient around the boundary of the focus.

– We observed a sharp potential around the boundary of the focus.

– When we induced two DSB’s we observed that there would exist one spherical focus, with a volume exactly two times larger than the focus for one DSB. This suggest that the two individual foci had fused together, which would be a fingerprint of two merging droplets.

– We found that by adding aliphatic alcohol hexanediol (a component proposed as a tool to differentiate liquid-like from solid-like assemblies), the Rad52 foci partially dissolved.

– Inside foci, Rad52 molecules exhibit confined motion, with a confinement radius matching with the size of Rad52 foci measured independently by PALM.

We have added details in the second-to-last paragraph of the introduction about the conclusions found in Min´e-Hattab et al., 2021.

Results:L73: maybe 'by the Boltzmann factor' is a bit easier to google for the uninitiated reader, rather than 'Boltzmann's law', which isn't really a thing.

We agree that this was a wrong formulation that could be confusing. We have corrected this to the “Boltzmann distribution”.

L76: 'While this description is general' is unclear, but could simply be left out.

We agree that this was an unnecessary formulation.

We have removed this, and changed it to “Here different…”

L80ff: Motivate why this potential is a good way to model binding sites? In the PMB model, the diffusion model for the binding sites ends up being very similar to the one used for LPM. Is it then fair to claim that this model is more "microscopic"? It seems that PBM is more microscopic only in words and not really in the modeling. One might say that the binding sites are just themselves following diffusion in an LPM. Then the difference between the two models is more about the fact that for LPM the tracer molecule diffuses in the LPM and in the other case the tracers do not feel directly the LPM but just bind to molecules (or binding sites) that are diffusing in the LPM. In this respect, the distinction between the two models does not contain much information on whether the foci are formed by phase separation or not. In view of the importance of this distinction and the ongoing controversies, the authors should clearly discuss this issue. To have a more microscopic model it might be useful to adopt a polymer model description, rather than a simple potential.

We start by emphasising the importance of the PBM with immobile binding sites (i.e. D_b_ = 0). It is typically assumed that the diffusion coefficient of the binding sites are significantly smaller than the diffusion coefficient of the free proteins, and this is also something we confirm in the paper Min´e-Hattab et al., 2021 (D ≈ 1.0µm^2^/s vs D_b_ ≈ 0.006µm^2^/s). Therefore one can in most situations approximate the binding sites as being static, and in this case the potential of the focus is not relevant, it is merely the position of the binding sites. Here we can assume that there is a region with high density of binding sites (i.e. inside the focus) and a region with no binding sites. We can then apply the techniques used in Figure 2 to derive the relation between the effective diffusion coefficient and the free energy. However, since the diffusion coefficient of the binding sites is non-zero we need to make sure that the theory did not break down by including this, and as soon as the binding sites are diffusing, one needs to introduce a potential in order to keep them inside the focus, since otherwise they would just spread over the entire nucleus. The potential we are introducing here is therefore merely an effective way to to maintain moving binding sites within the focus. In all realistic parameter ranges (i.e. D > D_b_) the binding/unbinding interactions with the binding sites, lead to the effective diffusion coefficient in a description that is also valid for diffusing binding sites. This model is therefore more microscopic since we use fundamental interactions to find the effective diffusion coefficient of the particles, whereas in the LPM it is a free parameter in itself that is not directly related to simple and fundamental interactions. We have added the following explanatory part to the discussion:

“To clarify this, we have removed the word “microscopic” in the section where we introduce the PBM, and we have added a sentence describing how the diffusion coefficient in this model is not a free parameter but an observable that we can measure based on the interactions with the binding sites. Furthermore, we have also removed the word “microscopic” from the Discussion section, and added a sentence stressing that the motion inside the focus is a result of the interactions with the binding sites.”

L92: Why exclude potential? Relevant for confinement, shape of droplet boundary (which might be a significant part of the droplet, for such small droplets). Also, six isn't much worse than five parameters.

We agree that this formulation leads to confusion, since we do not exclude the potential, we just don’t consider it as a parameter since the biding sites are kept inside the focus. Therefore we feel that this formulation should be changed.

To clarify this confusion we have changed the description following the equation 5 in the revised manuscript to emphasize that the only role of U_b_ is to keep binding sites within the focus, and is thus not fitted.

L105: Maybe say that parameter values are mentioned in Figure legend.

We agree that his is helpful to the reader and thank the referee for pointing this out. In the revised manuscript we have added: “(see Figure 2 caption)”

L109: Why just the simple Gaussian of free diffusion? Shouldn't there be confinement effects for small droplets?

For sufficiently small time steps, molecules inside the focus (that do not interact with the surface potential) experience a free diffusion. For longer time-steps more traces will have interacted with the surface potential and this leads to the confined diffusion.

To clarify these points we have specified that we consider traces within the bulk of the focus which are not interacting with the boundary for small time intervals.

L116: Would be good to have references for k- in yeast. At least ballpark. Otherwise also look at other k-.

There are not many results indicating the actual off rate of Rad52 binding sites. In one recent study they found the dissociation constant k_d_ for Rad52 to vary between two observed sites from 5.6 nM to 101 nM (Saotome et. al, 2018 iScience), however we do not know the number of molecules present in the study and cannot derive the actual unbinding rates for these results. However for many other receptors the actual unbinding rates can differ by orders of magnitude. We also argue that in the first part of the Results section, we show how we expect the dynamics to be if the value of k­_ is much smaller. In this case the diffusion coefficient inside the focus should not follow a normal diffusion translocation histogram but be the sum of two separate events. If we had found these different diffusion coefficients in the data, we could have made a more simple conclusion. Since we do not find this in the data, we use this as an argument to say that if the focus was really a PBM, the off rate has to be fast.

To make these numbers visible as well as the large range of values for the dissociation constant, we have added a sentence highlighting this and citing the paper by Saotome et al., in the section related to Table 1, where we discuss the parameter values.

L120: Does Rad52 ever appear in groups, e.g. dimers or multiple times within the same repair site?

Monomeric and multimeric forms of Rad52, have previously been observed in vitro (see for instance: Saotome et al., 2018, Structural basis of Homology-Directed DNA repair mediated by RAD52). In our previous paper Mine-Hattab et al., 2021, we suggested that both forms were found in the nucleus, but since we found one diffusion coefficient inside the focus, we do not think this alters our results here. If it was the case that more diffusion coefficients were present, we would in principle still be able to calculate the average diffusion coefficient, inside and outside the focus, and thus recover the results of Figure 2H to separate the two models.

L127ff: This entire section is a bit less clear than the rest. Maybe a slightly longer, more systematic exposition could help?

We agree that this argument could be explained better, and since it is closely related to Figure 2D, both the top and the bottom panel should be cited in this section.

In the revised version we have added some sentences and cited the findings in Figure 2D in order to guide the argument. Furthermore, we have added an introductory sentence in the section below, in order to make this transition more clear.

L131: Rebinding description is a bit vague, maybe elaborate a bit? Can this be connected to appendix 1?

Right after a particle is released, it will be close to the binding site from which it was just released. If the binding site is perfectly absorbing and the particle is released exactly at the boundary, it will be immediately recaptured. However since the binding sites are partially absorbing, there is a fixed probability that the particles will escape the binding site from which it was just released. In the paper we cite (Kaizu et al., 2014), they show that by including the rebindings, the effective off rate is significantly lowered but one can still assume it is a constant rate and can thus be treated as an effective k_off_ in the derivations. This simplifies the mathematics of the section, and let us derive the result that we validate in the simulations, which confirms that this assumption is valid for the system.

In the revised version of the manuscript, we have expanded the discussion of how rebinding events can still be modelled as a rate. This is included in the description just before the citation of Kaizu et al., 2014, which is just before the definition of equation 6.

Equation 7: Shouldn't the second ∝ be an equal sign?

That is correct and we thank the referee for noticing this It has been changed in the revised version.

141: How realistic is this limit? Doesn't it imply also fast diffusion? In other words, it seems like equation 6 gives many constraints on what your rate can be. Please comment. How well does the time scale separation hold? If it doesn't, many of the conclusions would have to be modified.

We thank the referee for pointing this out, since this could create a scepticism for many readers. This limit is realistic for many systems. Equation 6 gives a constraint in the value K_+_, which arise as κ 7→ ∞. This is the limit where we call the reaction “diffusion limited”. This is contrary to “rate-limited” reaction where κ is very small. However, reguardless of that, large binding rates for the tracked particle (given by K_+_ρ(r)), can be achieved by a large density of binding sites ρ(r). This is one of the fundamental assumptions in the PBM, consistent with our choice of parameters.

In order to discuss this issue, we have added two sentences in the beginning of this part, just before the introduction of equation 9, highlighting that the rates should be large and that the defining property is the large density of binding sites.

L152: Recall the definition of δ r. Is average from equation 10 removed?

δr is the experimental displacement observed during δt, which dr is an infinitesimal step. The average in Equation 10 is correct (otherwise there would be a noise term).

We have added additional descriptions of expression in line 181, in order to define the meaning of δr and clarify that hδr2i/(2dδt)is the maximum likelihood estimator of the diffusion coefficient in a small region of space.

L161: Not sure about D_0_, according to table this is only part of LPM? Also not mentioned inL93.

We are sorry about this confusion. D_0_ is the effective diffusion coefficient inside the focus. This can be measured in experiments or simulations for the PBM, but it is a function of the other parameters. Therefore it should not be inserted in the table for the PBM, since the other parameters are more fundamental and they lead to a resulting diffusion behaviour inside the focus.

We have added a sentence explaining that D_0_ is a parameter that can be extracted based on the simulations in the PBM but not an input itself. This explanation is added to the text just prior to the result of Figure 2H.

L164: Typo in unit for rho.

We thank for this observation.

It has been corrected in the revised version.

L167: Is this shown? Is this Figure 2d?

Yes this is indeed what we show in Figure 2D.

In the revised version we have added this reference for this sentence.

L181 and equation 13: Inconsistent use of δ vs d.

We are sorry about the confusion in this regard. δr is the measured displacement between two successive measurements. Due to the limits of optical resolution, in experiments we have a time between measurements that is short however not infinitesimal. Therefore we refer to δr, when extracting data from experiments (for instance to measure D˜(r) ). However it should be correct that this estimation we make in equation 13, is an approximation of this displacement and therefore we should change it to δr. To clarify this we have added the following sentence to the reviewed version:

“In order to correct for this and avoid confusion, we have changed the definition in equation 13, so *d* is replaced by a δ and the = is replaced by ~.

Equation 13: In measuring this in 3C, is the average r as a function of r computed from displacements starting at r or arriving at r?

This is an important question that should be specified clearly in the text. This is the starting position of r. However we note that we have tried to calculate this as a function of the midposition as well, but this does not make significant deviations to the main results.

We have specified this in the caption.

L164: Is it obvious that dr cannot be negative? What are the consequences of this? Is it true at the focus boundary?

This is a consequence of equation 13, where it is seen that if D_b_ = 0 then the expression reduces to:<dr>=dtD(r)r

It is not obvious when one thinks about the problem, but comes directly out the mathematical formulation. The consequence is that for immobile binding sites, the dynamics of the PBM looks like the dynamics in a region with no defined potential but with a radial diffusion coefficient, calculated by using the Ito interpretation, without including the spurious term. It should be noted that this is true everywhere – also around the boundary. We have tried to make this statement clearer by adding two sentences after this statement, clarifying the difference between the situation when D_b_ = 0 and D_b_ > 0.

L195: Why underestimate and how is this seen? Underestimate with respect to what ground truth? Overall the discussion of 3C, D should be extended and made clearer. It is one of the main signatures used to tell PBM from LPM so it should be very clearly discussed.

This underestimation is seen since the red curve in Figure 3D is significantly more negative than the two others, predicted only by the diffusion and the equlibrium distribution. The reason for this is that there is a defined connection between the diffusion coefficient and the equilibrium distribution in the PBM, but this is not necessarily the case for the LPM. In particular if we “know” the diffusion coefficient of the possible binding sites through other experimental investigations, one can directly see if the observed diffusion coefficient leads to the predicted behaviour around the boundary.

We have tried to highlight this more clearly. When referring to Figure 3D we have added a sentence about how the underestimation is seen. Also we have added a sentence after the sentence trying to explain why the PBM prediction results in a mismatch when comparing to the LPM. This sentence is inserted after the phrase ending on ”the wrong model”.

L197: Didn't you say above it was never negative? Now only for this value?

It can be negative if D_b_ > 0, see point 21.

This is now clarified after Equation (13).

L199: Can you expand this a bit for a clearer picture?

It is seen for macroscopic objects (for instance a water strider) that the surface pressure can lead to reflection. This is a direct result of the Laplace pressure which is given by ∆p = −γ∇·~n. Microscopically, this occurs due to the energy between interacting molecules in the liquid forms a surface, that counteract deformations in a droplet, for instance created by a macroscopic molecules that hit the surface. However for microscopic molecules, this does not lead to a deformation in the surface and therefore this Laplace pressure does not affect this system. We wanted to include this part, since many readers tend to think of surface tension in this way, and this is something completely different. However, we feel that it adds more confusion and does not clarify the text and we have therefore decided not to include this sentence.

This sentence has been removed in the revised version of the manuscript since we felt it added more confusion and did not add important information to the questions raised in this paper.

L210: Without confinement in shouldn't it be uniform in any dimension?

Yes this is true, but in two dimensions the angle is easy to define and understand. As we increase the number of dimensions, more angles appear which leads to much more tedious derivations.

We agree that this formulation was misleading and we have corrected this by moving the ”2 dimensions” to the ending of the previous sentence.

L250: Can you comment more on the Burger Purcell limit? How relevant is the discussion in the present context? If it is relevant, it should be expanded.

We believe this is important since the work of Berg and Purcell is a classic result that some readers will be familiar with. Therefore it is important to us to comment that our result is in accordance with this fundamental work in the relevant limits. It does not give any new insight to the result, but it might help a big part of the readers to understand and accept the result. Therefore we do not deem it necessary to expand on this since it does not affect our results directly.

To motivate this comparison more, we have added a sentence explaining that this bound has previously been shown to be relevant to gene regulation, citing the work of Gregor et al., 2007.

L263: Missing 'of'.

We thank the referee for noticing this typo.

This has been corrected in the revised version.

L269ff: This seems quite weak as a test, compared to the previous results (in particular 2H). The PBM clearly also has a minimum, how different are these for different parameter ranges? It also seems very hard to control focus size in vivo.

We agree that as an experimental test, this part is not as strong as the previous sections, mainly since it might be experimentally more difficult to carry out in reality. However, we still feel that it is an important part of the paper, since our analysis shows that if the binding sites are immobile (i.e. D_b_ = 0) then according the equation 21 the optimal focus size is r_f_ = 0. This means that it does not have a minimum, and even for relatively fast moving binding sites (remember that Figure 4C with D_b_ = 0.01µ^2^/s is twice the experimental value), the advantage of having the focus is really minimal.

This result shows another important element that is unrelated to the physical observables: If a focus is a PBM with approximately immobile binding sites, it is not possible to obtain an advantage in terms of enhancing the on-rates of specific molecules. On the contrary, it increases the search time significantly. Therefore this result suggests that if a focus should increase the on-rates (which would be the case for Rad52 and DNA repair) then a focus with immobile binding sites would never be the optimal structure and a liquid droplet would be much more effective. However, if the focus had the purpose of silencing genes, then it would make sense to increase the search time significantly, and here a PBM is a strong suggestion.

Regarding the focus size in vivo, we showed in the paper Min´e-Hattab et al., 2021 that in the presence of 2 DSB, one focus is formed with a volume approximately double in size. Therefore it might be difficult to construct a continuum of focus sizes but it should be possible to test this at a few different volumes.

We modified this argument in the revised paper. In the final part of the section, following Equation 21, we have changed the description so we do not use the phrase ”rule out the PBM” but on the contrary we write ”…identifying an optimal focus size would be an indication that the model is not a PBM”. Furthermore, we have added a sentence to the discussion, where we argue that this part is not the most optimal test to setup experimentally, but it might be important in the understanding of why foci could be one structure or the other.

L297: Using 'condensates' as a verb required rereading this sentence for clarification, maybe rephrase.

We have replaced that word by “concentrates”.

Figures:General: It would be helpful to have bold one-sentence figure caption summaries. Please make sure to always mention the equations that each panel refers to in the figure caption.

We thank the referee for this point, that we agree will help the reader.

In the revised version, we have included one sentence for all four figures in the main text, and we have added references to the relevant equations in the figure captions.

1: Panel B: might be good to also have shading for the background of the nucleus. Otherwise the boundary could be mistaken for a nuclear membrane happened to us at first glance (also compare Figure 4b).

We thank the referee for pointing this out. We also realized that it can be confusing in particular when comparing to Figure 4.

In the revised version of the paper, we have updated Figure 1 based on these suggestions.

2: Panel D: caption: are (bottom) and (top) reversed?

This is correct and we thank the referee for pointing this out. They have been corrected in the revised version of the paper.

Panel F: Caption: please clarify 'using displacement histograms'.

We acknowledge that this formulation was not clear. What we mean is that within a small radial segment we calculate all displacement, and use the maximum likelihood estimator D˜ = ‹hδr^2^›/(2dδt) to calculate the diffusion coefficient inside this radial segment.

In the revised manuscript, we have added a part to this caption, specifying how we obtain this the diffusion coefficient.

Panel G: Displacement is misleading for the x-axis label, how about 'r' or something like 'dist. from nucleus'.

This is true and we thank the referee for helping us clear this up.

This has been changed to “Radial coordinate” for the revised version of the paper.

Panel H: reorder legend, otherwise data point 'experimental observation' looks like legend.X axis has D_0_, shouldn't this be D^~^? Same in caption for H.To us this panel has the strongest evidence that the model of choice should be LPM,is this true? If yes, please clarify in text.

We thank the referee for pointing this out and we agree on this point. We have tried to move the ‘experimental observation’ label so it fits better. We also thank the referee for raising the question D˜ or D_0_. We will try to make this clearer in the main text, since we define D˜(r) as the spatially dependent diffusion coefficient, whereas D_0_ is the diffusion coefficient inside the focus. This we will make clearer in the revised version.

We furthermore agree that this panel is the strongest, which is also why it occupies extra space to highlight it. However we believe that the main result is the equation 11 that is tightly related to Figure 2H, where the relation is shown to hold.

In the revised figure, we have inserted the “experimental observation” as a legend, so the data point is still clear to observe. We have added the distinction between D˜(r) and D_0_ in the section regarding Figure 2H. Finally, we have added one sentence emphasizing the importance of this figure and the related equation 11, in the main text after introducing equation 11.

Caption for H: Could you refer to the figures/sections in Miné-Hattab where thepartitioning is calculated? After a quick skim we weren't able to find it.

This is calculated in the bottom part of the Results section, related to Figure 7, and use the input as we show in the table of all experimental values (Table S1).

To clarify this, we have added this information in the Materials and methods section, guiding the reader where to look in the previous paper. This has been added to the caption of Figure 2H and to the main text related to Figure 2H.

3: Panel B: What do the error bars refer to?

The error refers the standard deviation of the mean. We have added that information in the caption.

Panel C: Why the dip in the bottom panel? Equation 13 is said to never be negative? Is only D_b_ varied between the bottom and top panels? If D^~^ is kept constant, shouldn't the plateaus be the same?

From Equation 13 we see that as Db grows, the more negative the value for dr is (given that ∂_r_U(r) is positive). Therefore the dip stems from the fact that D_b_ has a large value. Here it is 20 times larger than the experimental value, and for values that we find realistic it will not be negative. However it is possible for fast diffusing binding sites, and this is what we wanted to show in this figure. This also explains the difference in the plateaus, since the value of D˜ is not the same between the figures, since D_b_ (but only D_b_) has been varied.

In the revised version we have tried to better explain this issue of when it can be negative depending on the value of D_b_, (text related to the description of Figure 3C).

Panel D: Caption: black instead of blue line?Are these plots purely based on equation (13)? Same as in C, how can this be negative?

We thank the referee for pointing this out. The explanation that the blue line can be negative is again, that if the PBM has strongly diffusing binding sites, then the value of dr around the boundary can be negative. However it still fails to predict how negative it is, since the LPM has more freedom to diffuse fast inside the focus.

In the revised version of the paper, this has been corrected. Furthermore, we have shifted the blue, and the yellow color to shades of green, since these are related to the PBM model

Reviewer #3:Membraneless condensates have recently become a central focus of the molecular and cellular biophysics communities. While the dominant paradigm for their formation, liquid-liquid phase separation (LLPS), has been well established in a number of cases for large, optically resolved droplets, there are significant concerns regarding the generality of this mechanism for smaller foci or puncta, and other mechanisms have been proposed to explain their formation. The problem is that it is very difficult to distinguish experimentally between these mechanisms for sub-optical resolution condensates. In this article, Heltberg et al., propose a novel method, based on the analysis of single molecule tracks, that allows discriminating between the liquid phase model (LPM) and one of the challenger mechanisms, the "polymer bridging model" (PBM). This method relies on the statistics of individual displacements – diffusion, radial displacements, angular changes – which are showed theoretically to exhibit different signatures for the two models. With realistic data this is sufficient to discriminate between the models: for instance in the case of double strand break foci (DSB), building on a recent work by some of the same authors, this article convincingly rules out the PBM in favor of the LPM. The author also investigate the influence on these two models on the search time to reach a specific small target – a commonly invoked role of condensates – and show that only the LPM substantially accelerates this, which could provide additional means to experimentally discriminate between the mechanisms, on top of the intrinsic interest of this finding.This article is a welcome addition to the literature in this field, as it will help clarify the nature of these condensates, in particular below the optical resolution. It is well-written, interesting and the conclusions are justified. I particularly appreciate the effort to employ simulated data that are realistic for actual experiments, which strengthens the claims of applicability. Some aspects of the data analysis and of the modeling, however, are insufficiently discussed and would need to be precised / expanded.1) The modeling is made under the assumption of thermal equilibrium, without further discussion. The authors should comment on why this is reasonable, in particular in view of the presence of active fluctuations and of chemical reactions in these condensates.

First of all, the experimental measurements are carried out after the formation of the foci, and the time of observation (tens of seconds) is small compared the lifetime of foci (tens of minutes). Therefore we can assume that these measurements are not affected by the effects of formation and disruption of foci. Secondly, the data extracted to compute the results of Figure 2 (in particular for Figure 2H) are not very sensitive to the active fluctuations, since we derive an average diffusion coefficient inside and outside of the focus as well as a free energy difference between an inside and outside level. It is indeed very likely that the soup of proteins that forms the focus is active, however Rad52 is not involved in chemical reactions at the timescales we are looking at may be considered passive. This is supported by our investigations of the experimental results, where we have not seen any statistical differences as a function of the time of measurements, and we have no reason to believe that active fluctuations affect the diffusivity of Rad52 on the observed timescales. Regarding binding sites, they may also diffuse actively along with the genome and chromatin, but we describe this by an effective description of the motion of Rad52 on short time scales, so that active effects are folded into an effective diffusivity (left as a free parameter).

We want to highlight this issue as well as present our arguments of why this description is valid for the experiments considered in this work. We have added text between Equation 6 and Equation 7 summarizing the arguments outlined above.

2) How is the diffusivity measured? Are these measures corrected for experimental error (e.g. using three-point estimators)?

Estimates of the diffusion coefficients in Min´e-Hattab et al., 2021 were obtained in different ways. Our main method is to generate the displacement histogram, and then estimate the number of different diffusion coefficients in the population based on likelihood fitting and KStesting. Then we take for all the traces, and find the ones that we are certain belong to the slowest diffusion coefficient. These traces are the ones in the focus, but by doing it this way, we are not vulnerable to the position of the boundary and to determine which are in the focus based on their position. Then we compute the MSD curves for this distribution of slowly diffusing molecules, and fit the diffusion coefficient based on a confined fit (which has a good p-value). This method is strong since we are fitting a slow diffusion population and typically can reject traces belonging to the fast diffusion coefficient. We also include the possibility of separating traces if the molecule goes from inside the focus to outside or the other way round. The alternative way we calculated the diffusion coefficient, was based on the microscopy data, where we “cropped” all the traces that could be visibly identified as being inside the focus. This method had the strength that we could visibly follow all traces, but the drawback that we could mistakenly identify molecules as being inside the focus, then they could be under or above the focus, as discussed in the section above. However both method yielded similar results. It is also based on these methods that we extract the size of the focus.

In order to clarify this important point, we have added two sentences in the caption to Table I, describing how the diffusion was measured in the experimental paper, and added a new paragraph about experimental measurements in Materials and methods. In addition, we have clarified in the caption of Figure 2F that we extract the maximum likelihood value of D˜(r) in each radial segment.

3) The conditioning of the averages should be discussed, e.g. in Equation 13: I assume that it is in the Ito convention? Similarly for the angle changes.

We assume that the density of the binding sites, follow a radial distribution, with no significant angular dependency. Thus the average displacement ‹dr› is computed as a function of the initial position of the particle and averaged over all initial displacements with similar radial positions. It is indeed formulated in the Ito convention, which is why the “spurious” term appear in the first term of the second line of equation 13.

To clarify that we are using Ito convention, we have stated that we are using Ito convention for this paper, just before the introduction of equation 1. We have furthermore clarified in the section related to equation 13 and the section related to the distribution of the angles that we use the initial position when calculating the difference between the two connected points.